# Mini-PCDH15 gene therapy rescues hearing in a mouse model of Usher syndrome type 1F

Maryna V. Ivanchenko [1], Daniel M. Hathaway[1,2], Alex J. Klein[1], Bifeng Pan [1], Olga Strelkova[2], Pedro De-la-Torre[2], Xudong Wu[1], Cole W. Peters[1], Eric M. Mulhall[1], Kevin T. Booth[1], Corey Goldstein[1], Joseph Brower [2], Marcos Sotomayor[3], Artur A. Indzhykulian [2] & David P. Corey [1] ✉

Usher syndrome type 1 F (USH1F), caused by mutations in the protocadherin-15 gene (*PCDH15*), is characterized by congenital deafness, lack of balance, and progressive blindness. In hair cells, the receptor cells of the inner ear, PCDH15 is a component of tip links, fine filaments which pull open mechanosensory transduction channels. A simple gene addition therapy for USH1F is challenging because the PCDH15 coding sequence is too large for adeno-associated virus (AAV) vectors. We use rational, structure-based design to engineer mini-PCDH15s in which 3–5 of the 11 extracellular cadherin repeats are deleted, but which still bind a partner protein. Some mini-PCDH15s can fit in an AAV. An AAV encoding one of these, injected into the inner ears of mouse models of USH1F, produces a mini-PCDH15 which properly forms tip links, prevents the degeneration of hair cell bundles, and rescues hearing. Mini-PCDH15s may be a useful therapy for the deafness of USH1F.

Advances in gene therapy have positioned adeno-associated viruses (AAV) as a ubiquitous delivery tool for treating monogenic inherited disorders[1–3]. Their low toxicity and immunogenicity minimize adverse reactions and they produce long-term transgene expression. Furthermore, various AAV serotypes and variants display broad but distinct tropisms that allow targeting to specific cell types[4–7]. Capsid design and discovery efforts have dramatically improved AAV transduction rates, particularly in neuronal tissues[1–8]. However AAVs remain limited by their capsid packaging capacity, which only allows transport of ~5 kb of genetic material. Elements such as promoters, polyadenylation sequences, and other regulatory elements further limit the size of transgenes that can be packaged. Dual and triple AAV approaches are options for circumventing this drawback; however they rely on co-transduction of multiple viral particles in the same cell, and on efficient intracellular recombination of the viral DNA or expressed proteins[9–11].

A less common approach is to use rational protein design to generate shortened versions of the protein of interest. This strategy has been employed to develop mini- and micro-dystrophins to treat Duchenne muscular dystrophy, based on the observation that large deletions in the dystrophin gene are generally well tolerated[12]. More recently, mini-otoferlin genes have been used to probe the role of the protein in hair-cell endocytosis in the cochlea[13,14], mini-*CEP290* genes have been evaluated for treating CEP290-associated Leber congenital amaurosis[15], and a "condensed" form of human tuberin (cTuberin) designed for treating tuberous sclerosis complex[16]. However a significant challenge for minigenes is the need to understand the protein structure and binding partners well enough to delete non-essential domains while preserving function, especially when function may include resisting large forces.

Over the past decade, we and others have elucidated the molecular structure, function, and interactome of PCDH15[17–28]. This protein is a large, atypical cadherin with 11 link-like extracellular cadherin (EC) repeats and a membrane adjacent domain (MAD12) preceding its single transmembrane domain[22,29]. Serving as the lower portion of the tip link in mature inner ear hair cells, PCDH15 binds to cadherin 23 (CDH23)[22,26] to convey force from sound stimuli to the mechanosensory transduction channels[30,31]. It also forms transient lateral and kinocilial links, critical for hair bundle development and planar cell

[1]Department of Neurobiology, Harvard Medical School, Boston, MA, USA. [2]Department of Otolaryngology - Head and Neck Surgery, Harvard Medical School and Massachusetts Eye and Ear, Boston, MA, USA. [3]Department of Chemistry and Biochemistry, The Ohio State University, Columbus, OH, USA. ✉e-mail: david_corey@hms.harvard.edu

polarity[17,21,25], and is expressed in retinal photoreceptors and in auditory cortex interneurons[32,33]. In humans, mutations in *PCDH15* can cause isolated deafness or Usher syndrome type 1 F (USH1F). Usher syndrome (USH) is the most common cause of deaf-blindness worldwide. It is both genetically and clinically heterogeneous. Patients with the most severe forms of Usher syndrome, known as type 1 (USH1), experience profound congenital hearing loss and severe balance deficits. Additionally, USH1 patients develop prepubertal retinitis pigmentosa (RP), which progresses to nearly complete blindness by the fifth decade[34,35]. USH1F represents roughly 3–11% of all Usher syndrome cases[36,37].

PCDH15 has three alternatively spliced C-termini: CD1, CD2 and CD3, but only the CD2 splice form is necessary for PCDH15 function in hair cells in mice[23,25]. *PCDH15-CD2* has a ~5.3 kb coding sequence, too large for an AAV capsid.

In this work, we used our structural knowledge of PCDH15 to eliminate 3–5 EC repeats while retaining PCDH15 function, in order to package the coding sequence into a single AAV capsid. After characterizing eight such mini-PCDH15 versions in vitro and optimizing AAV expression cassettes for inner ear hair cells, we tested their function in vivo in three USH1F mouse models. One version, mini-PCDH15 version 4 (mini-PCDH15-V4), demonstrated proper protein localization and remarkable rescue of hearing. Our data suggest that mini-PCDH15 proteins can be used for treating deafness in USH1F patients, and could be tested as a therapy to treat blindness. Furthermore, this work demonstrates that shortened versions of genes may be used to treat other forms of hereditary hearing loss for which the gene product is too large for AAV's packaging limit.

## Results

### Design of mini-PCDH15 constructs

To rationally engineer functional mini-PCDH15s that are small enough to fit within the AAV packaging limits, we relied on our previous studies of the atomistic structure of PCDH15 and the functional assessment of its EC repeats[18,19,27]. The 11 EC repeats in PCDH15 are similar but not identical in sequence or function. They have highly conserved and critically oriented $Ca^{2+}$-binding residues, which are essential in conveying force along the length of the extracellular domain[27,38]. To make a version of PCDH15 small enough to package into a single AAV, we developed a strategy to eliminate 3–5 EC repeats that are not essential for function, making a mini-PCDH15 about 25–40% shorter than the full length protein. PCDH15's binding partner CDH23 remains unchanged, so the tip link is expected to be 10–15% shorter overall (Fig. 1a).

Previous molecular dynamics simulations of a shorter chimeric PCDH15 EC1-2 + EC8-10 fragment indicated that artificial junctions (such as EC2 + EC8) would bind $Ca^{2+}$ and would be stable and mechanically resilient[38]. We designed eight mouse mini-PCDH15s (Fig. 1b) featuring various artificial junctions. All contained the EC1-3 repeats to facilitate both parallel dimerization and interactions with CDH23[18,27] and the EC11-MAD12 tail for dimerization and interaction with transmembrane proteins[19,28]. While the composition of the mini-PCDH15 middle region varied, four of the mini-PCDH15s (V1-V4) included the kinked EC9-10 repeats that can provide elasticity to the tip link and might be relevant for tip link assembly and function[38]. The other four mini-PCDH15s (V5–V8) did not include EC9-10. Of the eight engineered mini-PCDH15s, only three (V4 – 3756 bp, V7 – 3729 bp, and V8 – 3750 bp) were short enough to be packaged into AAV along with gene expression elements.

Next, we performed structural predictions of monomeric mini-PCDH15-V4, −V7, and −V8 extracellular domains (in the absence of MAD12) using the AlphaFold2 program developed by DeepMind[39,40]. AlphaFold2 uses a deep machine learning algorithm, incorporating physical and biological knowledge about protein structure and multisequence alignments, to predict protein structures without relying on previously solved structures or structural templates. The structural predictions made with or without known structural templates gave models with high confidence (median pLDDT ~95) that were nearly identical to each other (Fig. 1c). Although these models lack bound $Ca^{2+}$, they were similar to high-resolution X-ray crystal structures of PCDH15 regions that have been solved previously[18–20,24,27,38], suggesting that the mini-PCDH15s would fold properly and make bona-fide EC-EC junctions.

### Testing binding of mini-PCDH15s to CDH23 in vitro

We then carried out functional screens in vitro to determine whether the eight mini-PCDH15s behaved similarly to full-length PCDH15. With expression in HEK293 cells lacking N-cadherin and detection with immunofluorescence, we found that all eight versions were properly trafficked to the plasma membrane (Supplementary Fig. 1a). Using immunogold scanning electron microscopy labeling, we observed that their N-termini were located extracellularly (Supplementary Fig. 1b). To test whether they bound CDH23, we used a cell aggregation assay in which HEK293 cells lacking N-cadherin expressed either a mini-PCDH15 or CDH23 along with green or red fluorescent markers, respectively. When mixed, cells formed aggregates of mixed red and green cells if CDH23 and full-length PCDH15 or any of the eight mini-PCDH15 versions was expressed, but did not aggregate if cells expressed GFP or mCherry only (Supplementary Fig. 1c), suggesting retention of the heterotypic EC1-2 handshake interaction between mini-PCDH15s and full-length CDH23[27].

To evaluate mini-PCDH15 binding to CDH23 under load, we used the nanoscale pulldown technique (NanoSPD)[41]. Cells were transfected with three plasmids: MYO10 fused to GFP-NanoTrap, which pulls eGFP or eGFP-fused proteins to the filopodia tips[42], CDH23 fused with eGFP, and a PCDH15 version fused with mCherry. MYO10 trafficking of CDH23-eGFP resulted in green punctate labeling of filopodia tips. When PCDH15-mCherry binds to CDH23, it should be co-trafficked. Indeed, with full-length PCDH15 we observed red punctate labeling of tips, which colocalized with CDH23-eGFP. Labeling was absent if PCDH15 lacked EC1 and EC2 or if it bore the deafness mutations I108N or R113G[43,44] (Supplementary Figs. 2 and 3). Most but not all of the mini-PCDH15 versions were co-trafficked with CDH23, indicating binding (Supplementary Figs. 2 and 3). Overall, mini-PCDH15 versions V3, V4, V5, V6, V7, and V8 performed at levels similar to wild-type PCDH15 in the NanoSPD assay. Together, these assays suggest that all mini-PCDH15s go to the cell surface and they bind to CDH23 as well as full-length PCDH15.

### Testing dimerization of mini-PCDH15s in vitro

X-ray crystallography, negative stain, and cryo-EM have revealed that the PCDH15 extracellular domain forms a parallel dimer, binding at both EC2-EC3 and EC11-MAD12 regions[18–20,28]. Single-molecule force spectroscopy showed that dimerization is critically important in enhancing the lifetime of the bond to CDH23[45]. To test whether mini-PCDH15s dimerize as well as full-length PCDH15, we expressed C-terminal histidine-tagged mouse mini-PCDH15 extracellular domains in Expi293 cells. Proteins were purified, and their stoichiometry was analyzed with size-exclusion chromatography (SEC) coupled to multi-angle light scattering (MALS). The three shortest mini-PCDH15s (V4, V7 and V8) were chosen for dimerization tests, as these were most likely to be efficiently packaged in an AAV capsid. SEC-MALS analyses suggest that the three mini-PCDH15s tested are indeed dimers in solution (Supplementary Fig. 4). As expected, dimerization was lost when we introduced three mutations in the known EC2-EC3 and EC11-MAD12 binding interfaces[18–20,28]. These experiments suggest that at least these three mini-PCDH15 versions will dimerize in vivo, properly stabilizing the bond to CDH23 in the tip link.

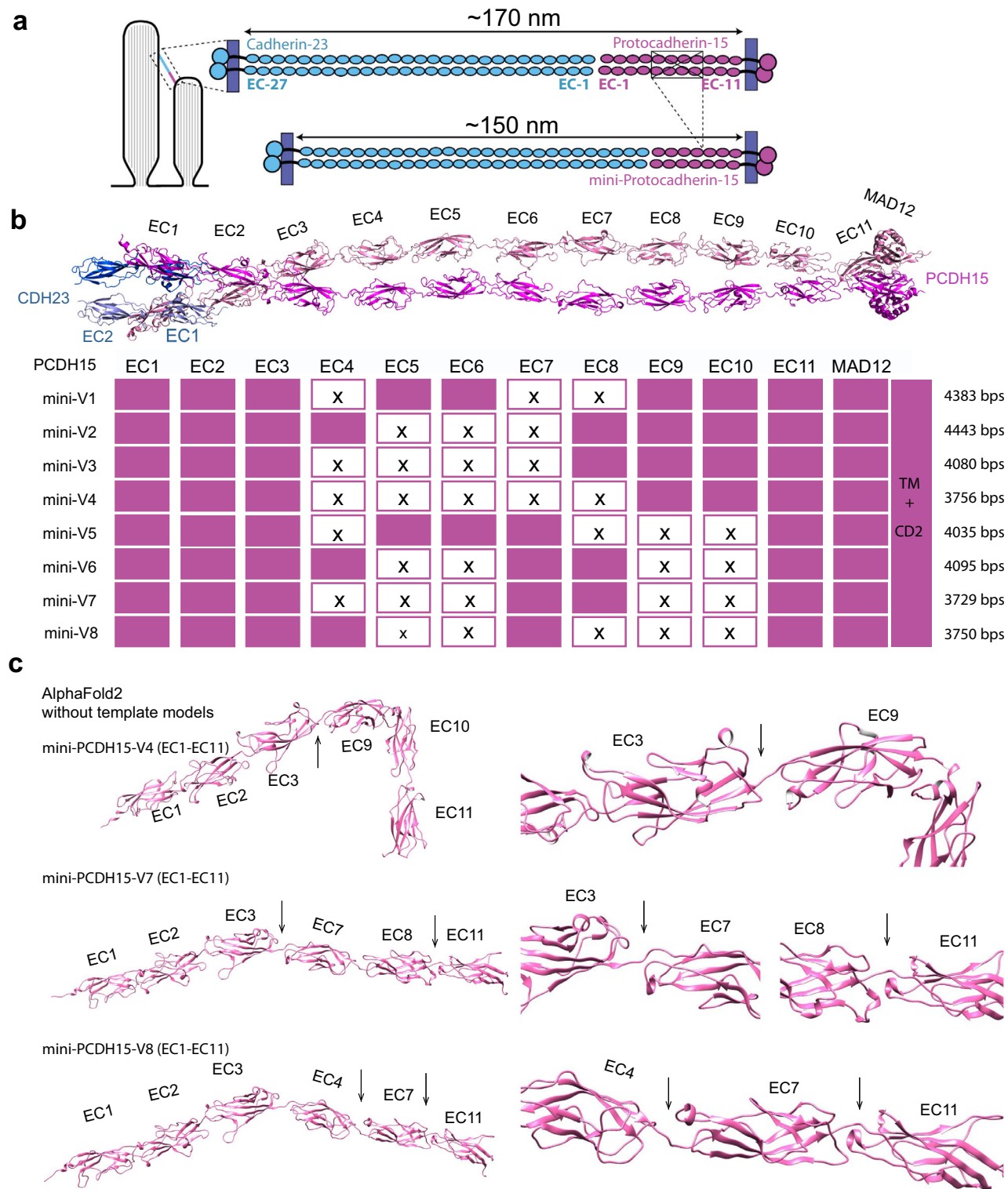

**Fig. 1 | Knowledge of atomic structures allows rational design of mini-PCDH15 constructs. a** Arrangement of the tip-link proteins PCDH15 and CDH23 at stereocilia tips. Each protein has multiple extracellular cadherin (EC) repeats forming a chain ~170 nm long. Mini-PCDH15s that lack 3–5 EC repeats are about 25–40% shorter, while the resulting tip links are 10–15% shorter. **b** upper panel, Composite atomic structure of wild-type PCDH15 (magenta) bound to EC1 + EC2 of CDH-23

(blue; adapted from Choudhary et al., 2020). lower panel, The EC repeats deleted from each mini-PCDH15 version. **c** AlphaFold2 monomeric structure models for the three shortest mini-PCDH15s (V4, V7, and V8), calculated without utilizing a structural template. The predictions with or without a structural template are nearly identical. Right panels are higher magnification images of the junctions between EC repeats (arrows).

## Optimization of AAV expression cassettes to maximize packaging capacity and transgene expression in the cochlea

Previous AAV gene therapy studies in the inner ear have used a variety of promoters and polyadenylation (poly(A)) sequences with or without a woodchuck hepatitis virus post-transcriptional regulatory element (WPRE)[46]. We generated four eGFP cassettes with different combinations of these elements to compare the transduction rate and expression levels and to define a minimal cassette suitable for expressing mini-PCDH15 (Supplementary Fig. 5). We tested the ability of these elements to drive eGFP expression by packaging them into AAV9-PHP.B, injecting them via the round window membrane (RWM) into cochleas of C57BL/6 J mice at age P1 (dose $1 \times 10^{11}$ VG), and evaluating eGFP expression in inner and outer hair cells (IHCs and OHCs) at P6.

In prior studies, we relied on a chicken beta-actin (CBA) promoter in combination with a cytomegalovirus (CMV) enhancer element and a chimeric intron for a total of 983 bp[5–7]. Here we compared this to the expression driven by a CMV promoter and CMV enhancer element but without the chimeric intron sequence, a total of 584 bp. Three different poly(A) sequences were also tested: a 434-bp SV40-BGH poly(A), a 234-bp BGH poly(A), and a 49-bp poly(A) sequence[47]. In addition, the 602-bp WPRE was also tested with each promoter using the SV40-BGH poly(A) sequence (Supplementary Fig. 5a).

We found that constructs containing the same WPRE and SV40-BGH poly(A) but different promoters (CBA vs. CMV) yielded differences in transduction rates and eGFP expression intensities in different hair-cell types (Supplementary Fig. 5b–d). We saw the broadest expression pattern with the CMV promoter. We next explored the impact on expression and transduction along the length of the cochlea, by varying the poly(A) and the presence of the WPRE. When the WPRE and SV40 poly(A) were removed, leaving the BGH poly(A), eGFP intensity sharply decreased in IHCs and OHCs, but the same fraction of cells were transduced. However, when the CMV promoter was used in conjunction with the 49 bp poly(A) sequence (without WPRE) both transduction rate and eGFP intensity dropped dramatically, with only the apical hair cells showing strong transduction (Supplementary Fig. 5b, c). Therefore we chose the CMV promoter in conjunction with the BGH poly(A) without a WPRE sequence to maintain high HC transduction rates while leaving space to package each of the three mini PCDH15 versions tested.

## Generation and characterization of conditional knockout mouse models for USH1F

Mice have previously been engineered to allow conditional deletion of exons encoding one or another of the three C-terminal splice forms (CD1, CD2 and CD3)[23,25], but each allows the possibility of compensation by another splice form. We therefore created a conditional deletion mouse model of USH1F deafness, $Pcdh15^{ex31}$, referred to here as $Pcdh15^{fl/fl}$, with loxP sites flanking exon 31 (Supplementary Fig. 6). Recombination by Cre recombinase deletes the critical transmembrane domain in all three C-terminal splice forms. We crossed this mouse with two different hair cell-specific Cre lines: $Gfi1$-$Cre$, expressed at E15.5[48], during the period of hair bundle development, and $Myo15$-$Cre$, expressed starting at P0 after the initial period of hair bundle development in the mouse inner ear[49].

Auditory brainstem response (ABR) and distortion-product otoacoustic emission (DPOAE) recording revealed that $Pcdh15^{fl/fl}$,$Gfi1$-$Cre^{+/-}$ mice were all profoundly deaf at P35 (Fig. 2a–c). Even at the highest sound intensity (120 dB SPL), these mice lacked identifiable ABR waves for all frequencies tested.

Scanning electron microscopy of conditional knockout cochleas at P6 showed that, as in null $Pcdh15^{Av3J}$ mice[50], the $Pcdh15^{fl/fl}$,$Gfi1$-$Cre^{+/-}$ hair bundles were significantly disorganized and misoriented compared to $Pcdh15^{fl/fl}$ control mice (Fig. 2d).

To test mechanotransduction in $Gfi1$-$Cre$ conditional knockout $Pcdh15$ mice, we used dye loading with FM1-43. This dye enters hair cells that have functional mechanotransduction and renders them brightly fluorescent[51]. At P10, FM1-43 loading showed no open transduction channels in IHCs and OHCs in $Pcdh15^{fl/fl}$,$Gfi1$-$Cre^{+/-}$ mouse cochleas, indicating no functional mechanotransduction (Fig. 2e).

Using immunofluorescence labeling with an antibody raised against a peptide epitope in EC1 of PCDH15 (DC811, see Methods; epitope identical to the well-characterized PB811 antibody[22]), a strong signal was detected at the tips of stereocilia of both IHCs and OHCs in $Pcdh15^{fl/fl}$ control cochleas (Fig. 2f). In contrast, and as expected, there was no labeling in either the disorganized and misoriented stereocilia or in the cell bodies of $Pcdh15^{fl/fl}$,$Gfi1$-$Cre^{+/-}$ mice at P6, both validating the antibody and confirming that exon 31 deletion eliminates PCDH15 (Fig. 2f). The DC811 epitope is extracellular, allowing us to perform immunogold scanning electron microscopy to detect PCDH15 on individual stereocilia. Using a field-emission scanning electron microscope with a backscattered electron detector, we found gold beads (12 nm) at the tips of stereocilia in $Cre^-$ animals (Fig. 2g), but not in $Pcdh15^{fl/fl}$,$Gfi1$-$Cre^{+/-}$ mice.

Deletion of $Pcdh15$ by the Cre recombinase under the $Gfi1$ promoter results in bundle defects in the early postnatal cochlea. We then generated a postnatal, hair cell-specific conditional knockout mouse model to delay these early morphogenetic defects, by crossing the floxed allele to $Myo15$-$Cre$ mice. We evaluated their auditory function with ABR and DPOAE at P35 and P60 (Fig. 3a–c, Supplementary Fig. 7). At P35 and P60, ABR thresholds in $Pcdh15^{fl/fl}$,$Myo15$-$Cre^{+/-}$ mice lacked any identifiable ABR response to loud sound stimulation across the frequency spectrum tested, indicating profound hearing loss (Fig. 3a; Supplementary Fig. 7). DPOAE thresholds at P35 had almost disappeared at all sound intensities and frequencies tested (Fig. 3b), and were completely absent at P60 (Supplementary Fig. 7b), indicating a complete loss of OHC function.

Cochlear hair bundle morphology from middle regions was analyzed with scanning electron microscope at P6 and P35 (Fig. 3d, e). At P6 in $Pcdh15^{fl/fl}$,$Myo15$-$Cre^{+/-}$ mice, no major abnormalities in bundle morphology were detected (Fig. 3d), unlike the bundle disorganization in $Pcdh15^{fl/fl}$,$Gfi1$-$Cre^{+/-}$ mice, which lack PCDH15 throughout development (Fig. 2f). Tip links were detectable (Fig. 3d). However, most middle row stereocilia had lost their normal shape, forming elongated tips, indicating actin core remodeling at the tips[52]. By P35 in the $Myo15$-$Cre$ mice, scanning electron microscopy revealed severely disrupted bundles: middle and short row stereocilia of IHCs and OHCs were shorter than controls or entirely absent (Fig. 3e), consistent with results showing stereocilia morphology changes dependent on mechanotransduction[52]. At P6, FM1-43 loading in $Pcdh15^{fl/fl}$,$Myo15$-$Cre^{+/-}$ hair cells was nearly similar to that in control mice. By P35, however, there was no detectable FM1-43 signal, confirming no functional mechanotransduction at later ages (Fig. 3f).

Overall, the Myo15-Cre mouse line is a good platform for screening mini-PCDH15 variants, because a late deletion of PCDH15 preserves hair-bundle structure well. This enables straightforward comparison of the ability of each mini-PCDH15 variant to form the tip link without the confounding presence of malformed stereocilia.

## AAV-mini-PCDH15s delivery rescues hearing in a $Pcdh15^{fl/fl}$, $Myo15$-$Cre^{+/-}$ mouse model

Having demonstrated a functionality of mini-PCDH15s in cell aggregation assays, NanoSPD, and dimerization tests; with an optimized vector construct for hair-cell expression; and with an appropriate mouse model, we went on to test different versions of mini-PCDH15 for efficacy in rescuing hearing in vivo.

We packaged in the AAV9-PHP.B vector[1,5,6,53] the three shortest mini-PCDH15 expression constructs: mini-PCDH15-V4, mini-PCDH15-V7, and mini-PCDH15-V8 (Fig. 4a). Expression was under the control

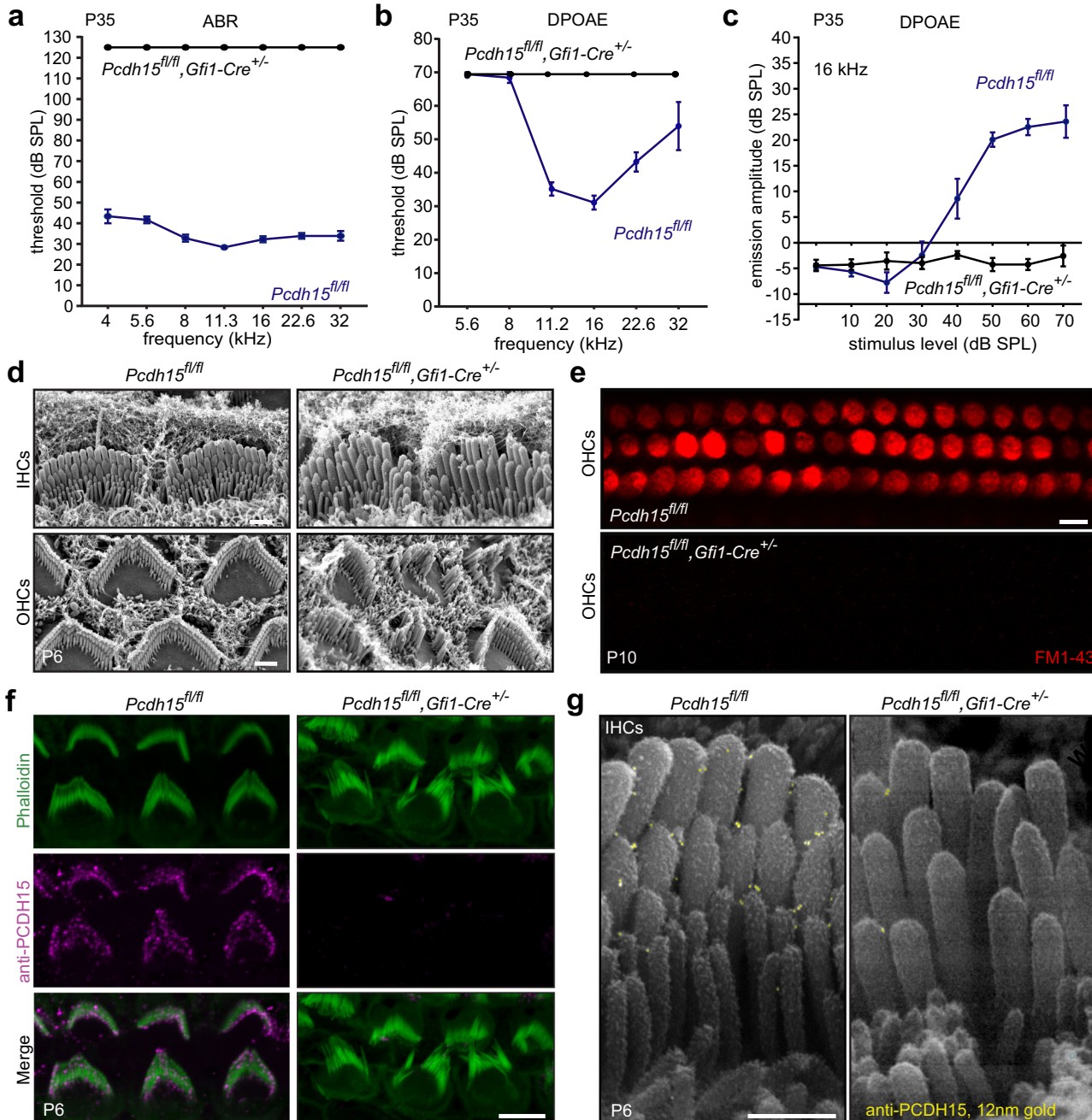

**Fig. 2 | *Gfi1-Cre* conditional *Pcdh15* knockout mouse cochleas show severe functional and morphological changes. a** Average auditory brainstem response (ABR) thresholds as a function of frequency in P35 mice. *Pcdh15^(fl/fl)^,Gfi1-Cre^(+/−)^* (*n* = 6) mice showed a profound hearing loss at P35 relative to *Pcdh15^(fl/fl)^* control mice (*n* = 9). **b** Knockout mice had no distortion-product otoacoustic emission (DPOAE) responses (*n* = 6) compared to *Pcdh15^(fl/fl)^* hearing control mice (*n* = 9). **c** DPOAE emission amplitude at 16 kHz in P35 *Pcdh15^(fl/fl)^,Gfi1-Cre^(+/−)^* (*n* = 5) was significantly reduced compared to the control mice (*n* = 4). **d** Representative scanning electron micrographs taken from P6 mice show disorganized inner hair cells (IHCs) (upper row) and outer hair cells (OHCs) (lower row) stereocilia morphology in *Pcdh15^(fl/fl)^, Gfi1-Cre^(+/−)^* hair cells (right) as compared to *Pcdh15^(fl/fl)^* controls (left). **e** Representative confocal microscopy images of FM1-43 dye loading by IHCs and OHCs from the apical region of the cochlea at P10. FM1-43 loading was completely abolished, indicating no functional mechanotransduction. **f** Anti-PCDH15 labeling (magenta) and phalloidin co-staining (green) in *Pcdh15^(fl/fl)^* control mice (left) demonstrated normal PCDH15 trafficking to stereocilia tips at P6, which was absent in *Pcdh15^(fl/fl)^, Gfi1-Cre^(+/−)^* knockouts (right). **g** Immunogold scanning electron microscopy labeling of a P6 IHC immunostained with anti-PCDH15 primary antibody and 12 nm gold-conjugated secondary antibody. PCDH15 trafficked to the tips of stereocilia in *Pcdh15^(fl/fl)^* control mice and was abolished in *Pcdh15^(fl/fl)^,Gfi1-Cre^(+/−)^* mice. Scale bars: (**d**) 1 μm; (**e**, **f**) 5 μm; (**g**) 0.5 μm. All data are presented as mean values ± SEM. Source data are provided as a Source Data file.

of the CMV promoter and BGH poly(A) sequences. All lacked five EC repeats, and all included the coding sequence for the CD2 C-terminal isoform of PCDH15. AAV vectors were injected via the RWM into *Myo15-Cre* conditional knockout mice at age P1, dose 5 × 10^10 GC. At age P35, treated animals were assayed for the rescue of hearing (Fig. 4b).

First, we examined whether the mini-PCDH15s could restore auditory function in conditional knockout mice by measuring ABRs in response to broadband clicks and tone bursts (Fig. 4c–e). At P35, the untreated mice were profoundly deaf, showing no detectable click ABR thresholds. Mice injected with AAV-mini-PCDH15-V8 were also profoundly deaf, indicating that version 8 could not substitute for full-

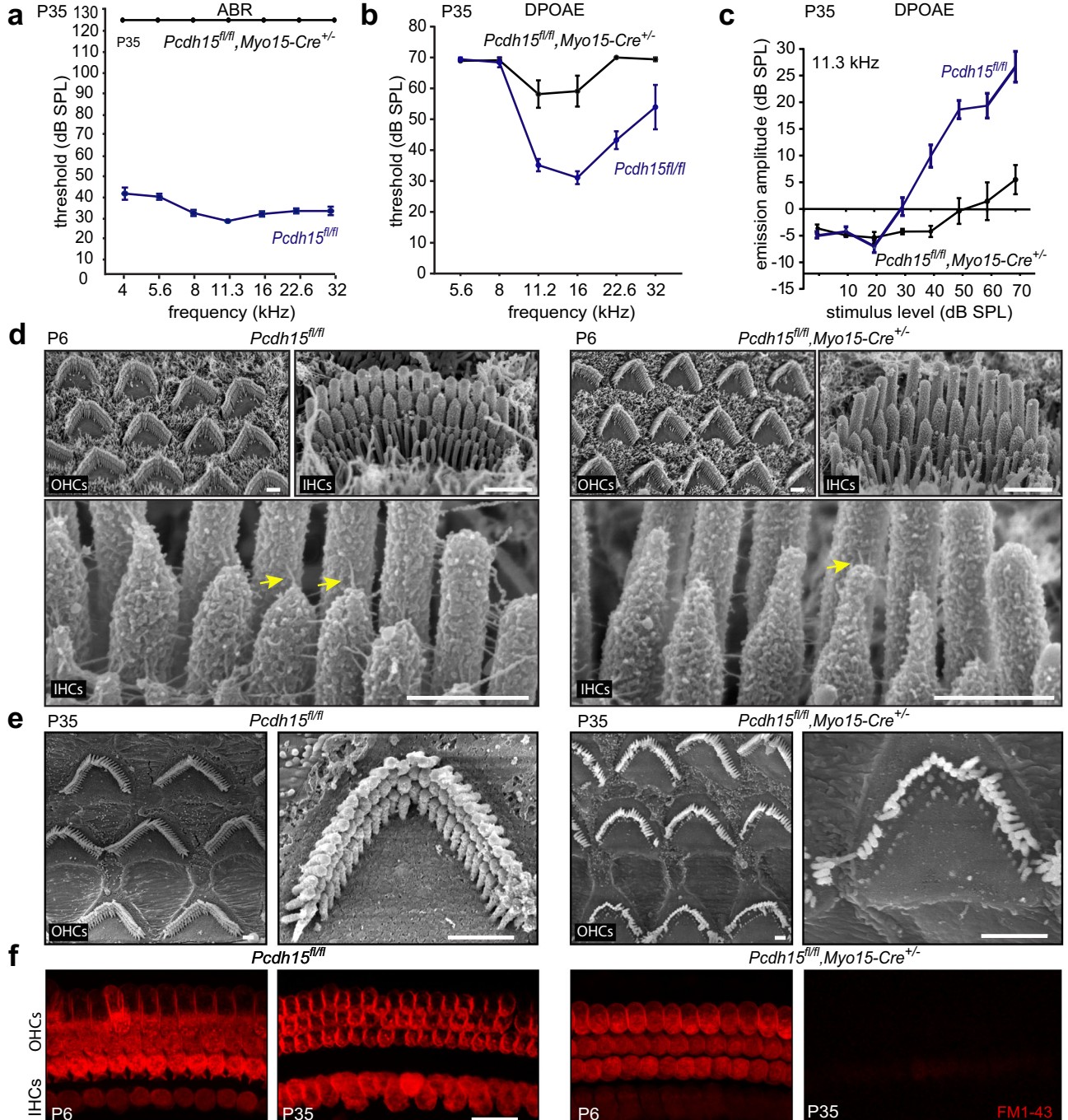

**Fig. 3 | *Myo15-Cre* conditional knockout mice show delayed functional and morphological cochlear pathology. a** P35, *Pcdh15^{fl/fl},Myo15-Cre^{+/−}* knockout mice (*n* = 10) show lack of auditory brainstem response (ABR) thresholds at all frequencies and sound levels tested, compared to *Pcdh15^{fl/fl}* hearing control mice (*n* = 9). **b** At P35, knockout mice (*n* = 6) have residual distortion-product otoacoustic emission (DPOAE) responses to high sound level stimuli in the mid-range frequencies, compared to *Pcdh15^{fl/fl}* control mice (*n* = 9). **c** Despite residual responses at high sound levels, DPOAE emission amplitude at 11.3 kHz in P35 *Pcdh15^{fl/fl},Myo15-Cre^{+/−}* mice (*n* = 8) was significantly reduced compared to *Pcdh15^{fl/fl}* control mice (*n* = 9). **d** Scanning electron micrographs of P6 stereocilia in *Pcdh15^{fl/fl}* hearing control mice (left) and *Myo15-Cre* conditional knockout mice (right) demonstrate healthy bundle morphology in neonatal knockout mice. High magnification images (lower panel) show extensive links between adjacent stereocilia (yellow arrows), including intact tip links in control mice as well as in knockout mice; however most middle row stereocilia in knockout mice had lost their normal shape, forming elongated tips. IHCs inner hair cells, OHCs outer hair cells. **e** Scanning electron micrographs of mature P35 stereocilia in *Pcdh15^{fl/fl}* control mice (left) and *Pcdh15^{fl/fl},Myo15-Cre^{+/−}* knockout mice (right) show that hair cells survive and the architecture of the organ of Corti is preserved at this age. However high magnification micrographs demonstrate shortened and missing middle- and short-row stereocilia in knockout mice. **f** In P6 mice, uptake of FM1-43 dye reveals a similar proportion of open transduction channels in *Pcdh15^{fl/fl}* mice and *Pcdh15^{fl/fl}, Myo15-Cre^{+/−}* mice. At P35, however, uptake was abolished in *Pcdh15^{fl/fl},Myo15-Cre^{+/−}* mice, indicating no open transduction channels. Scale bars: (**d**) (upper panel) 1 μm; (**d**) (lower panel) 500 nm; (**e**) 1 μm; (**f**) 20 μm. All data are presented as mean values ± SEM. Source data are provided as a Source Data file.

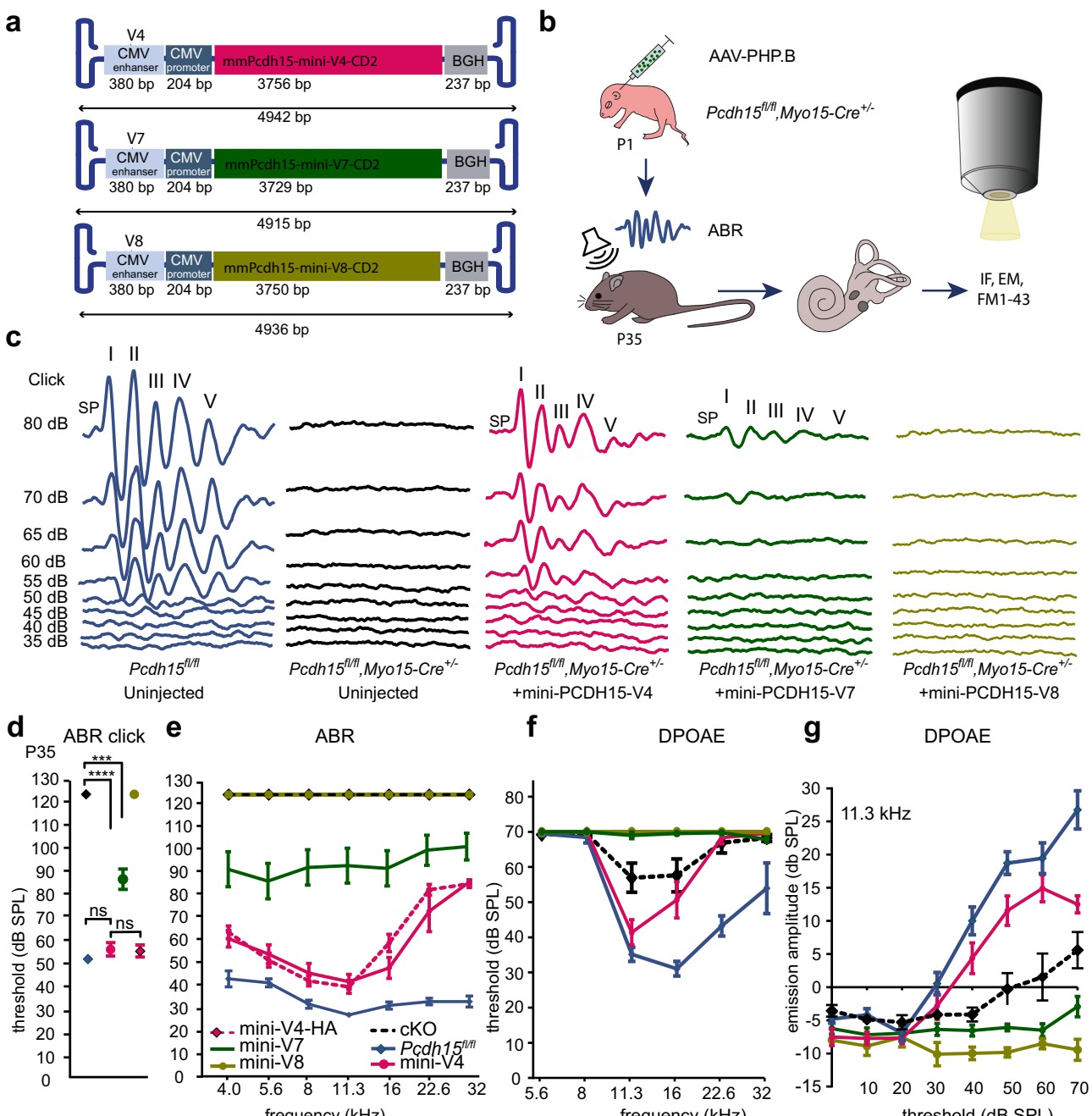

**Fig. 4 | AAV-mini-PCDH15s rescue hearing in the *Myo15-Cre* mouse model of USH1F. a** Schematic of the three mini-PCDH15 constructs that are short enough to be packaged into AAV capsids. The lengths of the DNA sequences are shown. Vectors used a CMV promoter and BGH polyadenylation sequence but not a WPRE. **b** AAV9-PHP.B was used to deliver the different mini PCDH15 versions at P1 via round-window membrane injection. IF immunofluorescence, EM electron microscopy. **c** Auditory brainstem response (ABR) testing showed differential rescue with different AAV-mini-PCDH15 versions. Representative click ABR traces from P35 *Pcdh15^fl/fl* hearing control mice, from uninjected *Pcdh15^fl/fl,Myo15-Cre^+/−* deaf knock-out mice, and from knockout mice injected with AAV-mini-PCDH15-V4, -V7, or -V8. SP, Summating potential. **d** Average click ABR thresholds for P35 uninjected *Pcdh15^fl/fl* control mice (n = 4), uninjected *Pcdh15^fl/fl,Myo15-Cre^+/−* conditional knockout mice (n = 6), and conditional knockout mice injected with AAV-mini-PCDH15-V4 (n = 7), AAV-HA-mini-PCDH15-V4 (n = 14), AAV-mini-PCDH15-V7 (n = 12),

or AAV-mini-PCDH15-V8 (n = 5). Significance was determined by a two-tailed unpaired *t*-test. *P* values: uninjected *Pcdh15^fl/fl,Myo15-Cre^+/−* vs. AAV-mini-PCDH15-V7, p = 0.0008 (***); *Pcdh15^fl/fl,Myo15-Cre^+/−* vs. AAV-mini-PCDH15-V4, p < 0.0001 (****); *Pcdh15^fl/fl* vs. AAV-mini-PCDH15-V4, p = 0.08 (ns); AAV-HA-mini-PCDH15-V4 vs. AAV-mini-PCDH15-V4, p = 0.84 (ns). **e** Average tone ABR thresholds as a function of frequency for P35 uninjected *Pcdh15^fl/fl* control mice (n = 9), uninjected *Pcdh15^fl/fl, Myo15-Cre^+/−* conditional knockout mice (n = 10), and conditional knockout mice injected with AAV-mini-PCDH15-V4 (n = 7), HA-tagged mini-PCDH15-V4 (n = 14), mini-PCDH15-V7 (n = 12), or mini-PCDH15-V8 (n = 5). **f, g** Average distortion-product otoacoustic emission (DPOAE) thresholds and average DPOAE emission amplitudes at 11.3 kHz, for P35 hearing control mice (n = 9) and uninjected *Pcdh15^fl/fl, Myo15-Cre^+/−* mice (n = 6), and for those injected with AAV-mini-PCDH15-V4 (n = 7), -V7 (n = 12), or -V8 (n = 5). All data are presented as mean values ± SEM. Source data are provided as a Source Data file.

length PCDH15. Mice injected with AAV-mini-PCDH15-V7 showed slightly improved click ABR thresholds ($87 \pm 6$ dB SPL), suggesting some functionality of version 7. However, mice injected with AAV-mini-PCDH15-V4 showed robust rescue of click-evoked ABR responses, measured as a threshold ($57 \pm 2$ dB SPL) near that of $Pcdh15^{fl/fl}$ control untreated mice ($51 \pm 1$ dB SPL) (Fig. 4c, d).

We also analyzed click-ABR wave-I amplitudes, representing the synchronous firing of auditory nerve fibers from the spiral ganglion neurons, and tested rescue with versions 4 and 7 (Supplementary Fig. 8). Overall, wave-I amplitudes in treated animals were reduced compared to $Pcdh15^{fl/fl}$ $Cre^-$ control mice. However, amplitudes were significantly higher in mini-PCDH15-V4- treated mice than in mini-PCDH15-V7-treated animals, and in some animals they reached hearing control levels (Supplementary Fig. 8).

We then measured sensitivity with tone-burst ABR recording at P35. Mice injected with AAV-mini-PCDH15-V8 were profoundly deaf, with no hearing rescue at any frequency tested. Mice injected with AAV-mini-PCDH15-V7 showed partial hearing rescue (Fig. 4e). However, the $Cre^-$ normal hearing control littermates injected with AAV-mini-PCDH15-V7 exhibited elevated ABR thresholds, suggesting that this version may be toxic (Supplementary Fig. 9). Finally, mice injected with AAV-mini-PCDH15-V4 showed near-complete rescue at low and middle frequencies and significantly improved hearing at high frequencies compared to untreated mice (Fig. 4e). Thresholds measured with tone bursts were generally more variable among animals than with click stimuli; in some animals, tone thresholds were as low as those of $Pcdh15^{fl/fl}$ mice (Supplementary Fig. 10). $Pcdh15^{fl/fl}$ $Cre^-$ control mice injected with AAV-mini-PCDH15-V4 had normal hearing at most frequencies, indicating little or no toxicity at P35 (Supplementary Fig. 9). The rescue persisted at P60 in treated $Myo15$-Cre conditional knockout mice (Supplementary Fig. 11).

Measurements of DPOAEs showed no rescue in mice treated with AAV-mini-PCDH15-V8 and AAV-mini-PCDH15-V7, and rescue in mice treated at P1 with AAV-mini-PCDH15-V4 (Fig. 4f). DPOAE amplitude measured at a representative midrange frequency (11.3 kHz) showed rescue to wild-type level in conditional knockout mice treated with AAV-mini-PCDH15-V4. In contrast, untreated mice and mice injected with versions 7 and 8 had no detectable DPOAEs (Fig. 4g). We tested toxicity in $Pcdh15^{fl/fl}$ control mice and found that mice injected with AAV-mini-PCDH15-V7 almost completely lost DPOAE responses, while mice treated with AAV-mini-PCDH15-V4 had normal DPOAE thresholds (Supplementary Fig. 9).

## Delivery of AAV-mini-PCDH15-V4 rescues stereocilia bundle morphology and mechanotransduction in the *Myo15-Cre* conditional knockout mouse model

Of the three mini-PCDH15 versions tested, mini-PCDH15-V4 showed the best hearing rescue measured by ABR. We therefore evaluated rescue of stereocilia bundle morphology and mechanotransduction with mini-PCDH15-V4, using both a fluorescent actin label and scanning electron microscopy. Staining of hair bundle actin with phalloidin showed that by P35, untreated conditional knockout mice had lost hair bundles or had severely disorganized bundles, with loss of short and middle row stereocilia and only some of the tall row remaining (Fig. 5a). On the other hand, injection of AAV-mini-PCDH15-V4 robustly rescued the morphology of hair cells, with bundles at P35 appearing like those in $Pcdh15^{fl/fl}$ control mice (Fig. 5a). Quantification of rescue of morphology (Fig. 5b) showed that $80 \pm 3\%$ of IHCs and $88 \pm 1\%$ of OHCs in the cochlea apex and $77 \pm 3\%$ of IHCs and $88 \pm 1\%$ of OHCs in the middle turn were fully rescued and were morphologically similar to control $Pcdh15^{fl/fl},Myo15$-$Cre^{-/-}$ bundles. Fewer bundles in the basal turn of the cochlea were fully rescued: $44 \pm 4\%$ of IHCs and $53 \pm 4\%$ of OHCs.

Scanning electron microscopy similarly showed well-organized hair bundles in IHCs and OHCs in $Pcdh15^{fl/fl}$ control mice (Fig. 5c). In

untreated $Pcdh15^{fl/fl},Myo15$-$Cre^{+/-}$ conditional knockout mice at P35, however, both IHC and OHC hair bundles were disrupted—showing a loss of short- and middle-row stereocilia—or had regressed entirely. In contrast, conditional knockout mice injected with AAV-mini-PCDH15-V4 displayed robust rescue of hair-bundle morphology of IHCs and OHCs and clearly visible tip links linking adjacent stereocilia (Fig. 5d–f).

To assess hair-cell mechanotransduction function in AAV-mini-PCDH15-V4 treated hair cells, we dissected cochleas out at P35 and briefly applied FM1-43 by local perfusion through the oval and round windows directly on the exposed epithelium. We quantified the proportion of FM1-43-positive cells in the apical and mid-apical regions of the cochlea (Fig. 5g). FM1-43 labeling at P35 demonstrated robust rescue of mechanotransduction in $86 \pm 3\%$ of OHCs and in $77 \pm 3\%$ of IHCs, compared to ~100% in normal hearing littermates (Fig. 5h).

## Mini-Pcdh15 transcripts are present in the cochleas of treated mice

The rescue of both hearing and bundle morphology by a vector encoding mini-PCDH-V4 strongly suggests that the transgene is properly expressed in the sensory epithelium. To confirm the presence of mini-Pcdh15-V4 transcript in treated mice, we isolated mRNA at P35 from control $Cre^-$ mice, from uninjected $Myo15$-Cre conditional knockout mice, and from organs of Corti in cochleas injected at P1 (Supplementary Fig. 12). Following reverse transcription, we amplified a PCR product with the size expected for full-length $Pcdh15$ cDNA (2002 bps) from $Pcdh15^{fl/fl}$ control samples, but it was absent in non-injected conditional knockout samples. In AAV-mini-PCDH15-V4 treated mice, RT-PCR demonstrated the presence of the $mini$-$Pcdh15$ transcript (388 bps) (Supplementary Fig. 12a, b). Amplicons from $Pcdh15^{fl/fl}$ control samples and AAV-mini-PCDH15-V4 injected samples were subcloned, and representative clones were subjected to Sanger sequencing (Supplementary Fig. 12c). Sanger sequencing confirmed the correct EC3-EC9 junction sequence in the $mini$-$Pcdh15$-$V4$ cDNA, which was absent in the control cDNA.

## AAV-expressed mini-PCDH15-V4 localizes properly at the tips of stereocilia

The rescue of hearing and bundle morphology also suggests that mini-PCDH15 is expressed and properly targeted in hair cells. To evaluate correct trafficking and localization of exogenous PCDH15 at the tips of cochlear stereocilia, we made a vector that includes a hemagglutinin (HA) tag at the N-terminus of PCDH15. The N-terminus of PCDH15 is a short helix extending away from the first EC repeat and away from the bond with CDH23, so the HA tag itself should not interfere with normal binding (Supplementary Fig. 13a). We first assessed protein production and localization in vitro. HEK293 cells were transfected with HA-tagged full-length PCDH15 and with untagged PCDH15 as a control. The cells were fixed and then immunolabeled with antibodies to PCDH15 or to the HA tag. We observed strong immunofluorescence labeling along the membranes with either anti-PCDH15 or anti-HA antibodies, indicating that the HA tag does not abrogate the trafficking of PCDH15 to the HEK cell surface (Supplementary Fig. 13b). We repeated these experiments with immunogold scanning electron microscopy labeling, which confirmed normal protein transport to the cell membrane and no apparent disruption due to the HA tag (Supplementary Fig. 13c).

To test localization of mini-PCDH15 in hair cells, we packaged HA-mini-PCDH15-V4 into the AAV9-PHP.B vector, using the same expression cassette as previously. First, we examined whether the HA-mini-PCDH15-V4 was able to restore auditory function in $Pcdh15^{fl/fl},Myo15$-$Cre^{+/-}$ conditional knockout animals. Thresholds of ABRs at P35 in treated mice confirmed that hearing sensitivity was rescued to the same level as with untagged mini-PCDH15-V4 (Fig. 4d, e).

Immunofluorescence imaging revealed that knockout mice injected at P1 with AAV-HA-mini-PCDH15-V4 ($5 \times 10^{10}$ GC) displayed

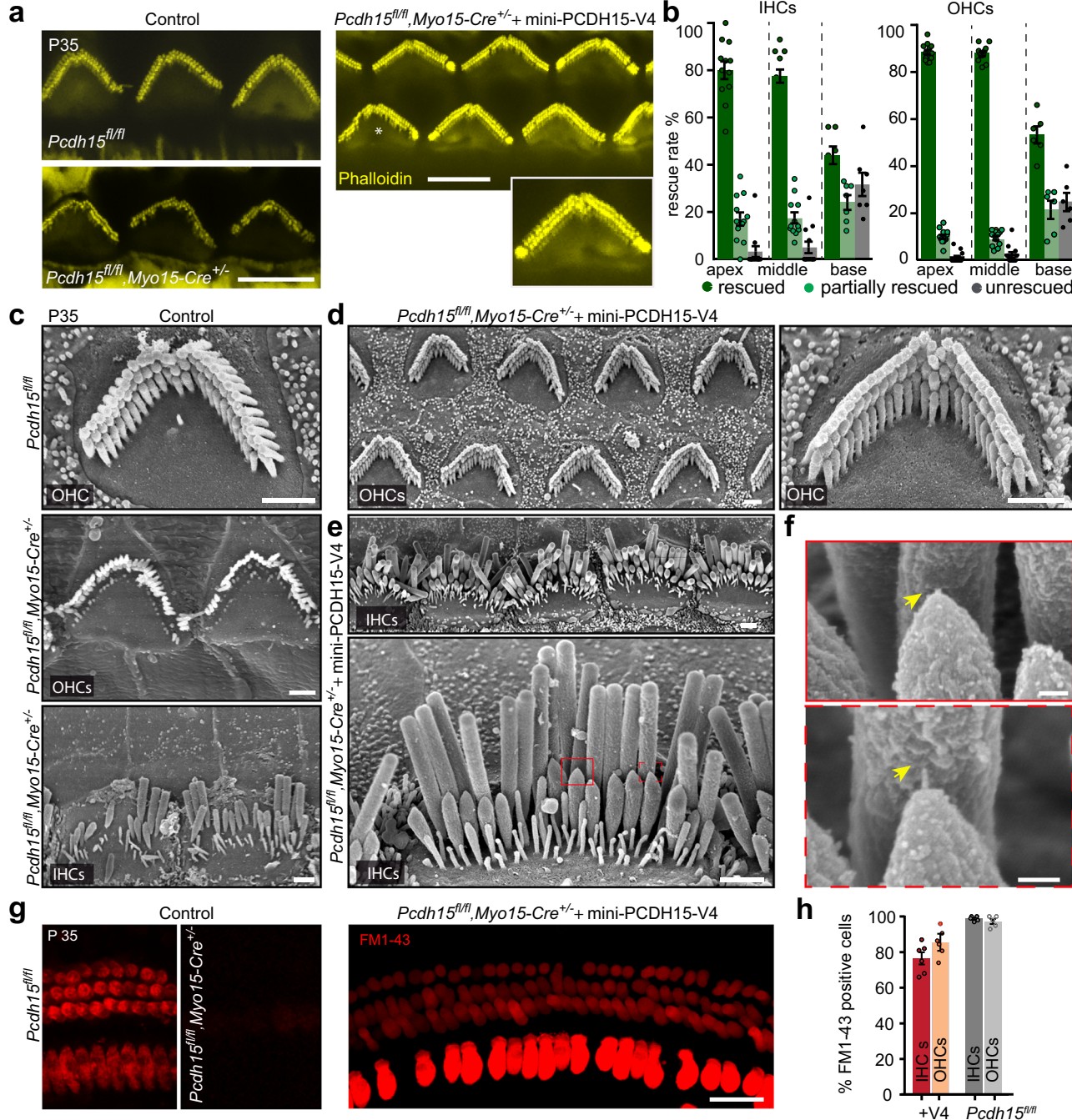

**Fig. 5 | Delivery of AAV-mini-PCDH15-V4 rescues stereocilia bundle morphology and mechanotransduction. a** Representative confocal microscopy images taken from the middle turn of the cochlea of P35 outer hair cells (OHCs) stained with phalloidin. Panels show *Pcdh15fl/fl* hearing control mice, uninjected *Myo15-Cre* conditional knockout mice or knockout mice injected with AAV-mini-PCDH15-V4. Phalloidin staining (yellow) shows rescued hair bundles in injected cochleas. Asterisk indicates a typical partially rescued hair bundle with >55% intact stereocilia per bundle. **b** Quantification of rescue rate of inner hair cells (IHCs) and OHCs in mice injected with AAV-mini-PCDH15-V4 in the apical, middle, and basal regions of the cochlea (*n* = 12 cochleas). Data are presented as mean values ± SEM. **c** Scanning electron micrographs of single hair bundles. Panels show an OHC of a hearing control mouse and OHCs and an IHC of an untreated conditional knockout. Knockout bundles are severely disrupted. **d** OHCs of a conditional knockout treated with AAV-mini-PCDH15-V4 showing normal bundle morphology. **e** IHCs of a conditional knockout treated with AAV-mini-PCDH15-V4 showing normal bundle morphology. **f** High magnification images of boxed areas in (**e**). Tip links were observed (yellow arrows) in rescued bundles. **g** Representative confocal microscopy images, from the apical/mid-apical region of the cochlea, of FM1-43 dye loaded IHCs and OHCs at P35. *Pcdh15fl/fl* mice (left*)*, uninjected knockout mice (middle), mice injected with AAV-mini-PCDH15-V4 (right). **h** Average percentage of IHCs and OHCs loaded with FM1-43 in conditional knockout mice injected with AAV-mini-PCDH15-V4 (*n* = 6), and in *Pcdh15fl/fl* control mice (*n* = 5). Data are presented as mean values ± SEM. Scale bars: (**a**) 5 μm; (**c**, **d**, **e**) 1 μm; (**f**) 100 nm; (**g**) 15 μm. Source data are provided as a Source Data file.

immunoreactivity to the HA tag. Label was properly located at the tips of stereocilia of cochlear IHCs and OHCs (Fig. 6a). Anti-HA labeling at P35 demonstrated high numbers of transduced cells throughout inner ears. Hair cell expression of HA-mini-PCDH15-V4 varied from apex to

base, with ~95–99% of apical and middle IHCs and OHCs transduced and ~68–75% of basal hair cells (Fig. 6b). Integrated fluorescence intensity of HA-tag labeling in bundles from the apical, middle, and basal regions showed variability among individual transduced cells.

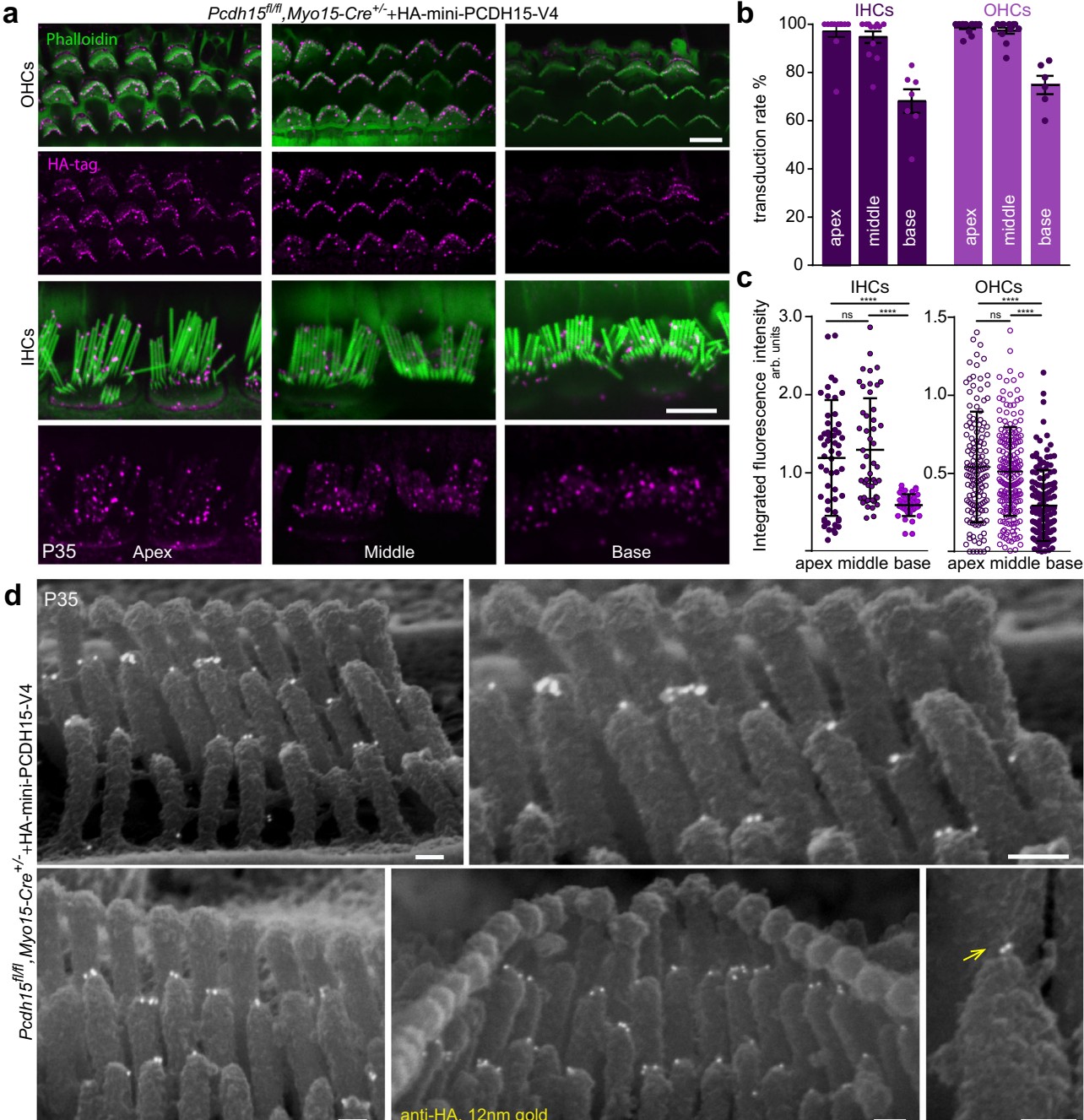

**Fig. 6 | Anti-HA staining of HA-tagged mini-PCDH15-V4 demonstrates localization at the lower end of the tip link. a** Representative confocal microscopy images at P35 from apical, middle and basal turns of the cochlea show anti-HA staining of HA-mini-PCDH15-V4 (magenta) at the tips of stereocilia (co-stained for phalloidin, green) in knockout cochleas treated at P1 with AAV-HA-mini-PCDH15-V4. IHCs inner hair cells, OHCs outer hair cells. **b** Transduction efficiency in IHCs and OHCs at P35 in treated *Pcdh15^fl/fl,Myo15-Cre^+/-* conditional knockout mice (*n* = 12). Data are presented as mean values ± SEM. **c** Quantification of fluorescence intensity of anti-HA labeling in IHCs and OHCs from apical, middle and basal turns of the cochleas in mice treated with AAV-HA-mini-PCDH15-V4. Symbols represent integrated fluorescence intensity in arbitrary units (arb. units) in individual cells (apex:

*n* = 56 IHCs, *n* = 144 OHCs; middle: *n* = 49 IHCs, *n* = 198 OHCs; base: *n* = 42 IHCs, *n* = 138 OHCs). Cells were measured from four separate cochleas. Data are presented as mean values ± SD. Significance was determined by a two-tailed unpaired *t*-test. *P* values: apex IHCs vs. middle IHCs, *p* = 0.45 (ns); apex IHCs vs. base IHCs, *p* < 0.0001 (****); middle IHCs vs. base IHCs, *p* < 0.0001 (****); apex OHCs vs. middle OHCs, *p* = 0.40 (ns); apex OHCs vs. base OHCs, *p* < 0.0001 (****); apex OHCs vs. base OHCs, *p* < 0.0001 (****). **d** Immunogold scanning electron microscopy demonstrated localization of HA-tagged mini-PCDH15-V4 at the tips of stereocilia (yellow arrow), similar to the localization of wild-type PCDH15 detected with anti-PCDH15, confirming that mini-PCDH15-V4 targets properly. Scale bars: (**a**) 5 μm, (**d**) 100 nm. Source data are provided as a Source Data file.

Fluorescence was not significantly different between apex and middle-turn regions, but was lower by about 50% in the base—consistent with the reduced hearing rescue at high frequencies (Fig. 6c).

Because the HA tag epitope is extracellular, we could then use immunogold labeling to detect HA-mini-PCDH15-V4 on individual

stereocilia. Using a field-emission scanning electron microscope, we observed gold beads (12 nm) on hair bundles, specifically on the short and middle row stereocilia at the position of the tip links (Fig. 6d), confirming proper trafficking and localization of HA-mini-PCDH15-V4 at P35.

### AAV-mini-PCDH15-V4 delivery rescues hearing in a *Gfi1-Cre* mouse model of USH1F

Mini-PCDH15-V4 rescues hearing in the delayed-deletion *Myo15-Cre* mouse model of USH1F. To test rescue in a model of USH1F with earlier deletion, we crossed *Pcdh15*<sup>fl/fl</sup> to *Gfi1-Cre* mice. The onset of *Gfi1-Cre* occurs in hair cells at E15.5 in the cochlea, coinciding with the generation of hair cells[48]. We found that early deletion of *Pcdh15* by *Gfi1-Cre* caused a more severe phenotype than in *Myo15-Cre* mice, with severely disorganized, fragmented, and disarrayed stereocilia bundles at P6 (Fig. 2d, f).

We asked whether the best performing mini-PCDH15 was also able to rescue hearing in *Gfi1-Cre* conditional knockout mice. The AAV-mini-PCDH15-V4 vector was injected through the RWM in P1 *Pcdh15*<sup>fl/fl</sup>,*Gfi1-Cre* knockout mice (dose $5 \times 10^{10}$ GC), ABR thresholds were tested at P35 (Supplementary Fig. 14). The uninjected mice were profoundly deaf, showing no detectable ABR responses for clicks or tone bursts up to 120 dB. However, mice injected with AAV-mini-PCDH15-V4 showed rescue of ABR, with hearing improved by ~50 dB (Supplementary Fig. 14). The rescue was best in low and middle frequencies and less but still significant at high frequencies, suggesting that rescue is possible even in cells with severe bundle disruption.

### Delivery of AAV-mini-PCDH15-V4 rescues stereocilia tip links, mechanotransduction and hearing in *Pcdh15*<sup>R245X/R245X</sup> mice

Although mini-PCDH15-V4 rescued hearing in the early-deletion *Gfi1-Cre* mouse model, there might be concern that this model does not adequately represent USH1F patients with congenital hearing loss. We therefore used a *Pcdh15*<sup>R245X</sup> constitutive null knockout line. The *R245X* mutation produces a stop codon early in the coding sequence, affecting all splice isoforms. It is the most common mutation in Usher 1F patients, so this mouse most faithfully mimics the human genotype. *Pcdh15*<sup>R245X/R245X</sup> mice, as previously reported for *Pcdh15*<sup>Av3J</sup>[50], were profoundly deaf at P35 (see below) and exhibited severely disorganized, fragmented, and abnormally polarized hair bundles at P6 (Fig. 7a, c). Scanning electron microscopy of P6 organ of Corti confirmed the absence of tip links in *Pcdh15*<sup>R245X/R245X</sup> animals compared to the regular arrangement of stereocilia and the presence of tip links in *Pcdh15*<sup>R245X/+</sup> mice (Fig. 7a). We confirmed the absence of PCDH15 protein with immunohistochemistry on P6 cochleas in homozygous *Pcdh15*<sup>R245X/R245X</sup> mice (Fig. 7c), and confirmed the absence of mechanotransduction using FM1-43 loading (Fig. 8a).

We then asked whether mini-PCDH15-V4 was also able to rescue the deafness phenotype in constitutive knockout mice. The AAV-mini-PCDH15-V4 vector was injected through the RWM in P0-P1 *Pcdh15*<sup>R245X/R245X</sup> mice (dose $5 \times 10^{10}$ GC). In mice treated with AAV-mini-PCDH15-V4 we observed rescue of tip links linking adjacent stereocilia in both IHCs and OHCs (Fig. 7b).

To evaluate proper trafficking and localization of mini-PCDH15-V4 at the tips of cochlear stereocilia and restoration of hair bundle morphology, we performed immunofluorescence labeling against the HA tag along with phalloidin staining. While no anti-HA signal was detected in the uninjected mice, strong immunolabeling was observed at the tips of cochlear IHCs and OHCs in *Pcdh15*<sup>R245X/R245X</sup> mice injected with AAV-HA-mini-PCDH15-V4 (Fig. 7d). Overall, anti-HA labeling at P6 demonstrated high numbers of transduced cells throughout inner ears injected with AAV-HA-mini-PCDH15-V4. In the organ of Corti, a majority of hair cells were transduced, with 88–91% of IHCs and 73–79% of OHCs expressing mini-PCDH15-V4 (Fig. 7e). Phalloidin staining showed that stereocilia in hair bundles were partially or fully restored (Fig. 7d). Despite the high transduction rate (Fig. 7e), quantification of rescue of morphology showed that only 6–18% of IHCs and 7–12% of OHCs in the cochlea were fully rescued and were morphologically similar to control *Pcdh15*<sup>R245X/+</sup> bundles (Fig. 7f). Many more were partially rescued (71–79% of IHCs and ~67–72% of OHCs) but showed some bundle fragmentation or disrupted polarity.

To examine individual stereocilia, we then used immunogold scanning electron microscopy labeling. At P9, injected *Pcdh15*<sup>R245X/R245X</sup> IHCs expressing mini-PCDH15-V4 demonstrated localization of gold beads at the tips and sides of stereocilia of stereocilia, confirming that mini-PCDH15-V4 targets properly in *Pcdh15*<sup>R245X/R245X</sup> mice (Fig. 7g).

Next, we assessed hair-cell mechanotransduction in the treated cochleas at P6. While FM1-43 loading was completely abolished in *Pcdh15*<sup>R245X/R245X</sup> uninjected mice (Fig. 8a), indicating that they lack activated transduction channels, the cochleas treated with AAV-mini-PCDH15-V4 showed rescue of FM1-43 loading in both IHCs and OHCs (Fig. 8b). At P6, $79.8 \pm 2.2\%$ of IHCs and $81.1 \pm 1.9\%$ of OHCs showed FM1-43 labeling, compared to ~100% in normal hearing heterozygous littermates (Fig. 8c).

To investigate the ability of mini-PCDH15-V4 to rescue the activation of mechanotransduction channels, we recorded whole-cell currents from IHCs of treated mice. Electrophysiological recordings were performed on mid-basal sections of cochlear explants at P10-P17 using a stiff glass probe to stimulate hair bundles. Peak inward current amplitudes of $290 \pm 33$ pA were recorded from *Pcdh15*<sup>R245X/+</sup> hearing control mice (Fig. 8d, e) and no current was detected in deaf control *Pcdh15*<sup>R245X/R245X</sup> mice (Supplementary Fig. 15). In *Pcdh15*<sup>R245X/R245X</sup> hair cells expressing mini-PCDH15-V4 we saw rescue of the current with peak amplitudes of $173 \pm 31$ pA; in the best-rescued cell, the current reached 352 pA. (Fig. 8d, e). To compare activation curves, we normalized the currents for each cell to peak current amplitudes evoked by a series of 16 bundle step deflections ranging from −175 to 1135 nm and plotted the activation curves (Fig. 8f). The shape of the activation curve and the bundle deflection at half activation in *Pcdh15*<sup>R245X/+</sup> control hair cells was similar to that reported previously in wild type mice using the same stimulation technique (https://www.ncbi.nlm.nih.gov/pmc/articles/PMC9278870/figure/F2/ Fig. 8f)[54]. However, the activation curve of *Pcdh15*<sup>R245X/R245X</sup> hair cells expressing mini-PCDH15-V4 was shifted slightly to the right compared to *Pcdh15*<sup>R245X/+</sup> controls, likely a consequence of poor probe coupling to disorganized and off-axis hair bundles.

The rescue of hair bundle morphology and restoration of mechanotransduction in mini-PCDH15-V4 treated *Pcdh15*<sup>R245X</sup> constitutive null mice suggested that mini-PCDH15-V4 could rescue cochlear function. To test this possibility, we measured ABRs at P35 in untreated mice and mice injected at P0-P1 (Fig. 8g, h). The uninjected mice were profoundly deaf, showing no detectable ABR responses up to 120 dB. However, mice injected with AAV-mini-PCDH15-V4 showed rescue of ABR, with hearing improved by up to 40 dB (Fig. 8g, h). The recovery was best at frequencies between 11.3 kHz and 22.6 kHz. Thus hearing restoration is possible even in *Pcdh15*<sup>R245X</sup> constitutive null mice, which mimic the genotype of the most common human mutation and have no endogenous expression of PCDH15 at any time.

## Discussion

We and others have made significant advances in gene therapy for deafness, primarily with gene addition therapy for small genes whose coding sequences fit in a single AAV vector[3,5–7,55,56]. However, some deafness genes—including many that cause Usher syndrome— encode large proteins with coding sequences that will not fit in AAV. More creative approaches must be developed to express therapeutic proteins in the target cells[57,58].

In this study we rationally engineered eight mini-PCDH15 genes that lack 3–5 EC repeats to shorten the coding sequence. The design of mini-PCDH15 constructs was based on our previously solved atomistic structures of PCDH15[18,19,27]. In all the mini-PCDH15s that we designed, EC1-3 and EC11-MAD12 regions were included to retain the parallel dimerization of PCDH15. Four of the eight minis retained the EC9-10 linker suggested to be key for PCDH15 flexibility and elasticity[18,38]. Middle EC repeats of the PCDH15 extracellular domain were deleted in various combinations based on their structural variability and protein

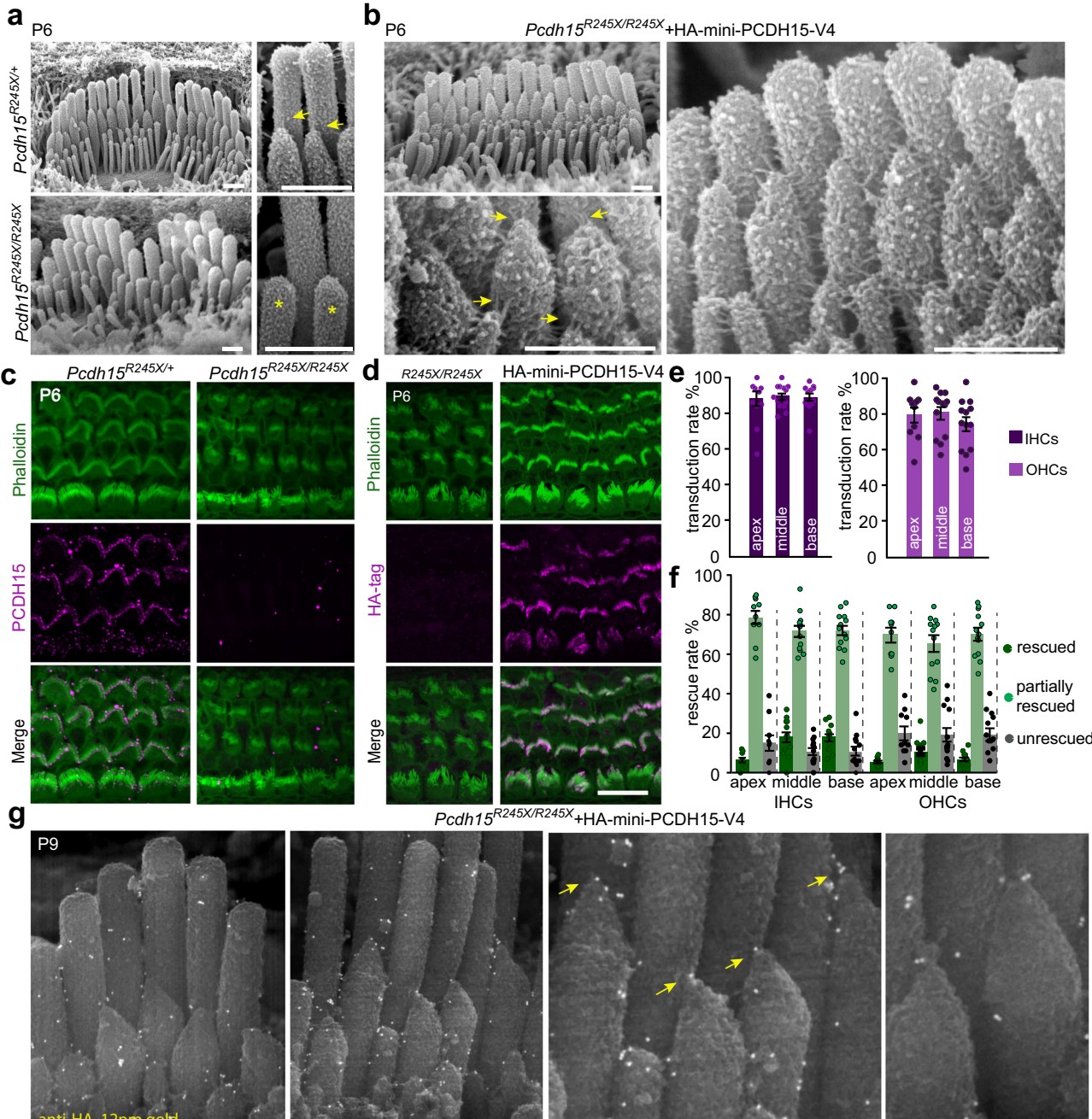

**Fig. 7 | Delivery of AAV-HA-mini-PCDH15-V4 rescues stereocilia tip links in the R245X knockout mouse model. a** Scanning electron micrographs of the organ of Corti from *Pcdh15^R245X/+^* hearing control mice at P6 (upper panel). Hair bundles have normal morphology and tip links connect the tips of adjacent stereocilia (yellow arrows). Organ of Corti from *Pcdh15^R245X/R245X^* homozygous null mice at P6 (lower panel). Bundles are severely disrupted, stereocilia are shortened, and no tip links are detected (yellow asterisk) although the lateral links are present. **b** Hair cells in *Pcdh15^R245X/R245X^* homozygous knockout mice expressing HA-mini-PCDH15-V4. Bundle morphology was partially or fully restored. High magnification images demonstrate the presence of tip links (yellow arrows) in rescued bundles at P6. **c** Representative confocal microscopy images of anti-PCDH15 labeling (magenta) of organ of Corti along with actin staining (phalloidin, green) demonstrated normal bundle morphology and PCDH15 localization to the stereocilia tips in *Pcdh15^R245X/+^* control mice (left). In *Pcdh15^R245X/R245X^* knockout mice the hair bundles were severely disorganized and no anti-PCDH15 labeling was detected (right). **d** Representative confocal microscopy images of the cochlea showing anti-HA staining (magenta). No

staining was observed at the tips of stereocilia in the untreated *Pcdh15^R245X/R245X^* mice (left). Anti-HA staining was observed at stereocilia tips in knockout cochleas treated with AAV-HA-mini-PCDH15-V4 (right). **e** Transduction efficiency in inner hair cells (IHCs) and outer hair cells (OHCs), measured with anti-HA labeling at P6, in *Pcdh15^R245X/R245X^* mice treated at P1 (*n* = 13). Almost all hair cells in all cochlear regions were transduced. Data are presented as mean values ± SEM. **f** Morphology rescue rate of IHCs and OHCs, assessed by fluorescent actin label, in mice injected with AAV-HA-mini-PCDH15-V4 in the apical, middle, and basal regions of the cochlea (*n* = 13). Most bundles were partially or fully rescued. Data are presented as mean values ± SEM. **g** Immunogold scanning electron microscopy of P9 IHCs expressing HA-mini-PCDH15-V4. Gold beads demonstrate localization of HA-tagged mini-PCDH15-V4 at the tips of stereocilia (yellow arrows) and along the surface of stereocilia, confirming that mini-PCDH15-V4 targets properly in *Pcdh15^R245X/R245X^* mice. Scale bars: (**a**, **b**) 500 nm, (**c**, **d**) 10 μm, (**f**) 200 nm. Source data are provided as a Source Data file.

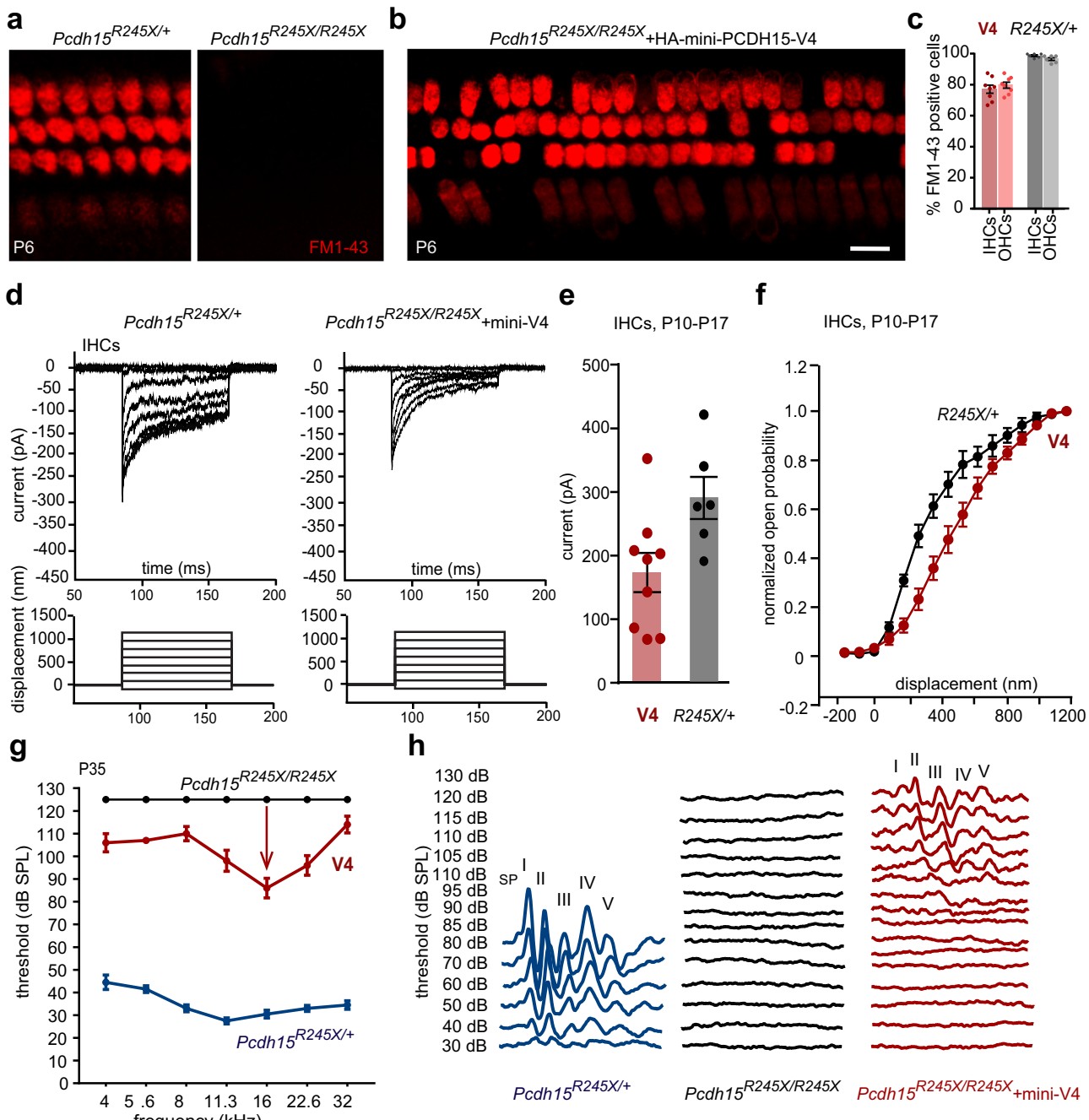

**Fig. 8 | Delivery of AAV-mini-PCDH15-V4 rescues mechanotransduction and the ABR in *Pcdh15^R245X/R245X* mice. a** Representative confocal microscopy images of FM1-43 dye loading in inner hair cells (IHCs) and outer hair cells (OHCs) from the middle region of the cochlea at P6. *Pcdh15^R245X/+* mice (left), uninjected *Pcdh15^R245X/R245X* (right). **b** FM1-43 dye loading in *Pcdh15^R245X/R245X* knockout mice injected with AAV-mini-PCDH15-V4. Loading was restored in most hair cells. **c** Average percentage of IHCs and OHCs loaded with FM1-43; knockout mice injected with AAV-mini-PCDH15-V4 (red, *n* = 8), uninjected *Pcdh15^R245X/+* normal control mice (gray, *n* = 6). About 80% of knockout cells loaded after treatment. **d** Representative transduction currents measured at P10-P17 (top) in response to −175 to 1135-nm bundle deflections (bottom). A *Pcdh15^R245X/+* hearing control IHC (left); a *Pcdh15^R245X/R245X* knockout IHC expressing mini-PCDH15-V4 (right). **e** Peak

transduction currents at P10-P17 from *Pcdh15^R245X/+* controls (*n* = 6 cells) and *Pcdh15^R245X/R245X* knockouts treated with AAV-mini-PCDH15-V4 (*n* = 9 cells). Cells from five separate *Pcdh15^R245X/+* cochleas and six separate *Pcdh15^R245X/R245X* treated knockout cochleas were studied. **f** Normalized open probability as a function of stimulus probe displacement for *Pcdh15^R245X/+* controls (black, *n* = 6 cells) and treated *Pcdh15^R245X/R245X* knockouts (red, *n* = 9 cells). **g** Average auditory brainstem response (ABR) thresholds as a function of frequency in P35 *Pcdh15^R245X/+* controls (blue, *n* = 10), uninjected knockout (black, *n* = 4) and knockout mice injected with AAV-mini-PCDH15-V4 mice (red, *n* = 5). **h** Representative ABR traces at P35 from controls, uninjected knockout mice, and treated knockout mice. SP summating potential. Scale bars: (**a**, **b**) 10 μm. All data are presented as mean values ± SEM. Source data are provided as a Source Data file.

sequence conservation[18], creating synthetic EC-EC junctions. As the strength of PCDH15 depends on $Ca^{2+}$ ions bridging acidic residues[27], the proper design of each synthetic junction was critical. The template-free structure prediction of monomeric mini-PCDH15s performed by the AlphaFold2 program gave models with high confidence (median pLDDT ~95), similar to structures of PCDH15 fragments solved previously with high-resolution X-ray crystallography[18–20,24,27,38] and to chimeric models with artificial EC-EC junctions[38], supporting the

expectation that mini-PCDH15s would retain similar structural conformation as the wild-type PCDH15 protein.

For this study, we chose the AAV9-PHP.B capsid for delivery because it transduces IHCs and OHCs in mouse and non-human primate cochleas and has the potential for clinical delivery[5,6,53]. To expand the size of transgenes packageable in AAV9-PHP.B, we explored options to minimize the size of the expression cassette to allow more space for the transgene. An expression cassette with a CMV promoter and BGH poly(A) sequence but no WPRE showed transduction and expression efficiency comparable to our previous vector[5,6] and left enough space to package all three mini-PCDH15 versions tested.

To test mini-PCDH15s in vivo, we generated two conditional knockout models. In both, exon 31—encoding the transmembrane domain and part of PCDH15's MAD12 domain—was deleted, but they differ in the onset of Cre activation. While both were deaf at P35, we found that early deletion by Gfi1-Cre resulted in stereocilia bundles that were severely disorganized at P6, near the time of AAV injection at P1. The hair bundles developed normally when the Cre was driven by the late-onset hair cell-specific Myo15 promoter, allowing us to test and differentiate mini-PCDH15 variants functionally.

The three shortest mini-PCDH15s—versions 4, 7, and 8—were packaged into AAV9-PHP.B and injected at P1 into Myo15-Cre conditional knockout mice. Two versions restored hearing function in vivo. Intriguingly, treatment with AAV-mini-PCDH15-V4, which had only one synthetic EC junction (EC3-EC9) and retained the flexible EC9-10 junction, showed strong hearing rescue and no toxicity, with rescue durable for at least two months. AAV-mini-PCDH15-V7, which had two synthetic junctions (EC3-EC7; EC8-EC11) and no EC9-10 repeats, showed only partial rescue and caused severe hearing loss in Pcdh15fl/fl control mice. Similarly, the non-functional mini-PCDH15-V8 had two synthetic junctions (EC4-EC7 and EC7-EC11) and no EC9-10 repeats.

Perhaps the presence of two synthetic EC-repeat junctions in mini-PCDH15-V7 and mini-PCDH15-V8 affects parallel dimerization mediated by EC1-3 and EC11-MAD12, or perhaps synthetic junctions between some EC repeats were not properly stabilized by $Ca^{2+}$. It is also possible that the flexible EC9-10 linker contributes to the proper assembly, elasticity, and function of mini-PCDH15 tip links and they are not stable without it. Structural studies of these mini-PCDH15s could provide insight for engineering even more efficacious versions.

Scanning electron microscopy and immunofluorescence further verified rescue by mini-PCDH15-V4 at P35. Nearly all hair cells (80–88%) maintained normal hair-cell bundle morphology in apical and middle regions. Furthermore, mini-PCDH15-V4 formed tip links properly located at the tips of short and middle row stereocilia, as clearly visualized with immunofluorescence microscopy and with conventional and immunogold scanning electron microscopy. Interestingly, even though the expression of eGFP with AAV9-PHP.B was relatively high in the base, the rescue level was highest in the apical and middle regions of the cochlea and significantly lower in the base, suggesting that the efficiency of eGFP expression did not directly correlate with the efficiency of PCDH15 expression.

We first used a late-onset Myo15-Cre mouse model, which allowed normal hair cell morphology at the time of mini-PCDH15 treatment and before endogenous PCDH15 deletion, as a platform for evaluating the relative efficacy of the different mini-PCDH15 versions. We then evaluated the best performer in a mouse model with earlier Pcdh15 deletion. In Pcdh15fl/fl,Gfi1-Cre+/− mice the Pcdh15 deletion occurred before complete hair bundle development. This mouse model had severely disrupted hair bundles similar to those described previously in null Pcdh15Av3J mice[50]. Nevertheless, the treatment with the best mini-PCDH15 version showed rescue of ABR, with threshold improved by ~50 dB. This is encouraging and suggests that treatment of USH1F deafness caused by milder mutations (e.g. R134G, G262D and V528D)[44,59] may be treatable with mini-PCDH15s.

To further validate the clinical relevance of this approach, we sought to test it using a mouse line bearing a mutation carried by USH1F patients. In the Ashkenazi population, a single nonsense variant (p.R245X) accounts for 64% of cases, so we used a Pcdh15R245X constitutive knockout mouse line, which mimics the human genotype. Mice were profoundly deaf and they had no expression of PCDH15 at any time, which led to severe disruption of hair-bundle morphology, absence of tip links, and the complete loss of mechanotransduction.

Treatment of Pcdh15R245X constitutive knockout mice at P0-P1 with AAV-mini-PCDH15-V4 partially rescued the phenotype. The tip links were clearly detected in treated P6 mice. Immunofluorescence and immunogold scanning electron microscopy labeling further confirmed the proper trafficking and localization of mini-PCDH15-V4. We were able to restore the mechanotransduction current and FM1-43 loading in cells expressing mini-PCDH15-V4. Finally, the evaluation of ABRs showed encouraging hearing rescue, with ABR thresholds improved by up to 40 dB. Delivery of a functional coding sequence at P0 was not sufficient to restore bundle morphology, however, so the incomplete rescue likely reflects the performance of well-functioning tip links on poorly organized stereocilia.

While AAV injection in the mouse inner ear may provide insights into potential therapies for congenital hearing loss, it may not necessarily translate directly to humans. The human inner ear develops on a different timescale relative to birth than that of mice, and there may be differences in the optimal timing, dosage, and delivery method of AAV vectors. The window of therapeutic efficacy for humans may close one week prior to hearing onset at 18 weeks gestational age, based on the postnatal window of efficacy observed in mice[60], and intervention beyond this critical period may not be as effective in restoring hearing. However, some mild mutations (e.g., R134G, G262D, and V528D) are not deleterious to balance and vision and are associated with late-onset hearing loss, which can potentially be treated[44,59]. In addition, vestibular hair cells, in particular, may be more responsive to therapy, as they continue to differentiate even after birth in mice[60]. Thus, early intervention targeting a subset of vestibular hair cells may be possible. However, the most promising initial therapeutic target for Usher 1 F is progressive blindness. Fortunately, the progression of blindness in Usher 1 F occurs gradually over several decades, providing a sizable window of opportunity for treatment[34,35,61,62].

Altogether, testing a mini-PCDH15 approach in three different mouse models helped us better understand how the timing of mini-PCDH15 treatment—before PCDH15 deletion, after PCDH15 deletion, or when no PCDH15 is expressed—affects the rescue level, providing more relevant data for their clinical translation in patients with USH1F. Mini-PCDH15 may also be a viable therapeutic option for addressing blindness. Finally, this study demonstrates that minigenes might be used to treat other forms of hereditary hearing loss for which the coding sequence is too large for AAV's packaging limit.

## Methods
### Study approval
All animal studies were conducted in compliance with ethical regulations according to protocol IS00001452 approved by the Institutional Animal Care and Use Committee (IACUC) at Harvard Medical School, Boston, and were performed according to the NIH guidelines.

### Study design
This study used structural knowledge of PCDH15 to engineer mini-PCDH15 versions in which 3–5 of the 11 extracellular cadherin repeats were deleted, but which still bound CDH23 (the other tip-link protein) and rescued hearing in mouse models of Usher 1 F. We narrowed eight candidates to three based on in-vitro assays, and tested three mini-PCDH15 versions in mouse models of Usher 1 F. Two Usher 1F mouse models were newly characterized and in three mouse models the therapy was tested. AAV vectors were injected in vivo, and the

outcomes were evaluated using RT-PCR, immunolocalization and confocal microscopy, conventional and immunogold scanning electron microscopy, FM1-43 loading, single cell electrophysiology, and measurement of ABR and DPOAE. Left ears were injected and uninjected mice served as controls. The study did not take into account the sex of the mice used. Initial experiments indicated no sex-specific hearing phenotype, therefore male and female animals were utilized in the experiments.

## Mini-PCDH15s design

The human coding sequence for PCDH15 CD1-1 (NM_001142763.1:396-6248) was used to generate initial versions of the mini-PCDH15 extracellular domains without exon 12a. Splitting of EC-EC junctions was done at DXNDN motifs or equivalent according to sequence alignments and structural models[18]. Corresponding mouse sequences using the CD2 isoform (based on NM_001142742.1) were generated. These synthetic templates were used for further cloning into various vectors as specified below. Detailed protein sequence information of the variants is illustrated in the Supplementary Table 1.

## Structure predictions

Structural models for the monomeric full length PCDH15 (EC1-EC11) and for mini-PCDH15 versions 4, 7 and 8 were generated by AlphaFold2[40]. The predictions with or without utilizing a structural template were performed in a Colab notebook (ColabFold v1.5.2, Google Research). To obtain multiple conformations, we ran the script multiple times. The top five of the ranked structures were analyzed. Structures were aligned in and visualized using Chimera v1.17 and PyMOL.

## Mouse models

Animal handling, breeding, and all procedures were performed in compliance with NIH ethics guidelines and under a protocol approved by the Animal Care Committee of Harvard Medical School. Mice were housed and bred at a Harvard animal facility. The *Pcdh15*$^{fl/fl}$ mice lacking Cre recombinase had hearing sensitivity and bundle morphology similar to wild-type (*Pcdh15*$^{+/+}$) mice, so in this study *Pcdh15*$^{fl/fl}$ *Cre-* mice were used as normal hearing controls. All studies were performed on *Pcdh15*$^{fl/fl}$,*Myo15-Cre* or *Pcdh15*$^{fl/fl}$,*Gfi1-Cre* mice on mixed C57BL/6J–129/Sv genetic backgrounds. *Myo15-Cre* mice[49] were a generous gift from Dr. Christine Petit (Institut Pasteur) with help from Dr. Ronna Hertzano (University of Maryland School of Medicine). *Gfi1-Cre* mice[36] were kindly provided by Dr. Lin Gan (University of Rochester). Genotyping for *Myo15-Cre* or *Gfi1-Cre* mouse lines was done as previously described[48,49]. *Pcdh15*$^{R245X/+}$ mice had hearing sensitivity and bundle morphology as in wild-type mice, so in this study *Pcdh15*$^{R245X/+}$ mice were used as normal hearing controls for rescue experiments in *Pcdh15*$^{R245X/R245X}$ mice.

## Generation of conditional knockout mice

A targeting vector was designed in which *loxP* sites were introduced upstream and downstream from *Pcdh15* exon 31, which encodes part of MAD12 and the single PCDH15 transmembrane domain (Supplementary Fig. 6a). Deletion of exon 31 occurs in all three C-terminus splice forms (CD1, CD2 and CD3). PCDH15 protein, bearing a signal sequence but lacking the transmembrane domain, is expected to be secreted. A neomycin resistance (neo) cassette flanked with FRT sites was introduced upstream of exon 31. The targeting construct was injected into embryonic stem cells. Positive stem cells carrying the intended construct were injected into blastocysts to obtain chimeric mice. After germline transmission, mice were crossed with 129/Sv mice expressing Flp recombinase to remove the neo cassette.

Genotyping primers for the *Pcdh15*$^{fl/fl}$ floxed allele were designed to detect the *loxP* sites (Supplementary Fig. 6). *Pcdh15*$^{fl/fl}$ mice were crossed with *Gfi1-Cre* or *Myo15-Cre* mouse lines. *Gfi1* expression begins at E15.5 in the cochlea, coinciding with the generation of hair cells[48].

*Myo15* expression in the cochlea starts at P0 at the base[49]. All studies were performed on mixed C57BL/6J–129/Sv genetic backgrounds. *Pcdh15*$^{fl/fl}$,*Myo15-Cre*$^{+/-}$ mice and *Pcdh15*$^{fl/fl}$,*Gfi1-Cre*$^{+/-}$ bred well and appeared healthy.

Generation of the *Pcdh15*$^{R245X/R245X}$ constitutive null knockout line will be described in detail elsewhere. Briefly, in *Pcdh15*$^{R245X/R245X}$ mice the mouse sequence: 5′-gaccgtgcacaaaatctgaatgagagag**c**gaacaaccaccacca ccctcacagtagatgttc-3′ was replaced with the sequence: 5′-gaccgtgcc-caaaatctgaatgagagg**t**gaaccaccaccaccactctcacagtggatgttc-3′, with a T > C substitution at position 250 that converts an Arg codon to a stop early in the coding sequence. Because the mouse models the human mutation at position 245, we refer to the mouse model as R245X rather than R250X. Mice were genotyped by amplifying the target locus using two primer pairs, cPcdh15f: 5′-ggagactggagggcagcaatcag-3′ and cPcdh15r: 5′-cattctgagacaggacttcagtggg-3′, and performing an XcmI digest, which produces a 300 bp band for the R245X allele.

## AAV vector constructs and pcDNA3.1 plasmids

The AAV expression cassettes tested in this study are shown in Fig. 4 and Supplementary Fig. 5. We used an AAV transgene plasmid, flanked by AAV2 inverted terminal repeats (ITRs). The mini-PCDH15 AAV expression constructs contained a 584-bp cytomegalovirus (CMV) promoter, the mouse *mini-Pcdh15* gene (CD2 isoform), and the bovine growth hormone (BGH) poly(A) sequence (Addgene deposition number – 82408).

To evaluate efficiency of gene expression regulatory elements, AAV plasmids carrying a single-stranded eGFP cassette were synthesized. Elements tested included the CMV promoter, a hybrid cytomegalovirus immediate-early enhancer/chicken beta-actin (CBA) promoter, a woodchuck hepatitis virus post-transcriptional regulatory element (WPRE), SV40 poly(A), a BGH poly(A), and a 49-bp poly(A). AAV plasmids ITRs were validated by running Sma1 digestion and transgenes and were Sanger-sequenced before packaging.

For evaluation of transgene expression in HEK293 cells, pcDNA3.1 mmPCDH15-CD2-IRES-EGFP, pcDNA3.1 HA-mmPCDH15-CD2-IRES-EGFP, pcDNA3.1 mm.mini-PCDH15-CD2-IRES-EGFP, were used. All plasmids were Sanger-sequenced before use and analyzed with SnapGene v 5.1.

## Viral vector production

AAVs generated in this study were produced by the Viral Vector Core at Boston Children's Hospital. Serotype AAV9-PHP.B vectors were packaged using HEK293T cells, by polyethylenimine-mediated co-transfection of pAAV transfer plasmid, pHelper plasmid, and RepCap plasmid pUCmini-iCAP-PHP.B. The media and cells were harvested 120 h post-transfection. AAV9-PHP.B viruses were released and subjected to discontinuous density iodixanol (OptiPrep, Axis-Shield) gradient ultracentrifugation. AAV vector-containing iodixanol fractions were removed after ultracentrifugation and concentrated by diafiltration. Purified AAV vectors were titered by Q-PCR. Vectors were pipetted into single-use aliquots and stored at −80 °C until use. They were thawed just before use for in vivo injections.

## Transfection of HEK293 cells

HEK293 cells (ATCC, #CRL-1573) were grown in six-well plates on glass coverslips in DMEM supplemented with 10% FBS (Gibco), and penicillin/streptomycin (Pen/Strep; Invitrogen). On the next day, cells were transfected with Lipofectamine 2000 following the manufacturer's protocol. Plates were incubated at 37 °C for 24 hrs and then at 30 °C for an additional 72 h.

## Cell aggregation assay

HEK 293 cells form lateral adhesions using an endogenous cadherin, CDH2, and exhibit strong calcium-dependent aggregation[63]. To avoid a potential artifact in testing PCDH15/CDH23 binding, we used 293NC

cells, a modified line which lacks CDH2 and does not form aggregates. An experimental protocol was adapted from Yamagata et al. [63] using their CRISPR/Cas9 N-cadherin (cdh2) knockout HEK293T cells (kindly provided by Dr. Joshua Sanes, Harvard University).

Briefly, transfected adherent cells were washed with Hanks' Balanced Salt Solution (HBSS) supplemented with calcium and magnesium (Gibco) before being trypsinized in 0.05% trypsin-EDTA (Gibco) for 10 min at 37 °C. Cells were resuspended in Dulbecco's Modified Eagle Medium with 10% FBS (DMEM, Gibco) before being centrifuged at $600 \times g$ for 5 min, then subsequently resuspended in DMEM supplemented with 0.5% Roche Blocking Buffer and 0.5% Bovine Serum Albumen (BSA). To test heterotypic interactions, $5 \times 10^5$ cells of each population were mixed and plated on an Ultra Low Cluster 24-well plate (Corning). Cells transfected with *Cdh23* were co-transfected with an mCherry expression vector whereas those transfected with *Pcdh15* expressed eGFP under an IRES to enable identification when cells were mixed. Cells were placed on a shaker at 100 rev/min for 12 h at 30 °C and then observed on an Olympus upright FV1000 confocal microscope equipped with a $60 \times 1.1$ NA water-dipping objective lens.

## NanoSPD assay

HeLa cells (ATCC, #CCL-2) were cultured on 18 mm coverslips in DMEM media supplemented with 10% FBS and penicillin-streptomycin antibiotic and incubated at 37 °C, 5% $CO_2$. Transfections were made with Lipofectamine 3000 (Thermo Fisher Scientific) using a standard protocol. Cells were fixed 28–30 h post-transfection in 4% formaldehyde for 15 min. Following several washes in PBS, coverslips were stained with CF405 phalloidin (Biotium) to visualize actin filaments and filopodia, supplemented with DAPI to visualize the nuclei. Samples were imaged on a Leica SP8 (Leica Application Suite X (LAS X) 4.0.2) scanning confocal microscope using a 63x, 1.3NA objective lens and a pixel size of ~90 nm.

To reliably detect red and green fluorescence above background, several image analysis procedures were used. First, all Z stacks were merged into a maximum intensity projection image using ImageJ. Next, for each cell, up to ten regions of interest were placed on puncta located on filopodia. An additional region of interest (ROIs) was selected to include a region of the coverslip without any cells to measure the background fluorescence. Within each cell, preference was given to puncta that appeared to have the strongest eGFP and mCherry signals; puncta located at the tip, along the filopodia or at their base were all included in the analysis.

Using ImageJ, up to 10 puncta were selected as individual ROIs, along with one ROI outlining the background fluorescence. The eGFP and mCherry fluorescence intensities were calculated for each ROI using the multi-measure function. All measured ratio values were plotted in GraphPad Prism, and the statistical significance was evaluated using multiple comparisons nested one-way ANOVA, and corrected for multiple comparisons using Tukey test. Overall, 605 cells and 5878 puncta were analyzed across all conditions from seven independent transfection experiments, each containing a subset of construct combinations. Each construct combination was evaluated at least in three independent transfection experiments.

## SEC-MALS experiments

Expression of mini-PCDH15 extracellular domains in Expi293 cells (Thermo Fisher, A14528). The coding sequences of mouse mini-PCDH15 versions V4, V7 and V8 extracellular domains were subcloned into NdeI and XhoI sites of a vector. For each construct, the native signal sequence is located at the N-terminal end, and a double-hexahistidine tag was inserted at the C-terminal end of the protein sequence. Suspension Expi293F cells were cultured in Expi293 medium at 37 °C, 8% $CO_2$, and incubated on an orbital shaker at 125 rpm[19]. Cultures (30 mL) were prepared at densities of $3.5–5.0 \times 10^6$ cells/mL and exchanged into ~29 ml of fresh and pre-warmed Expi293 media (37 °C) containing 0.5X penicillin/streptomycin in a 125 mL vented flask. For transfection, the DNA complexes were prepared by diluting 60 μg of DNA up to 1.5 mL with Opti-MEM using 15 mL conical tubes. Separately, 240 μg of polyethylenimine hydrochloride (PEI, MW 40,000 − 1 mg/mL stock) was diluted up to 1.5 mL with Opti-MEM. The PEI mixture was added into the DNA solution and immediately vortexed for ~10 s and incubated for an additional 20 min at room temperature. The complexes were then added to the cells and incubated for ~20 h at 37 °C. Next, 35 μL of 100X penicillin/streptomycin, 40 μL of sodium butyrate 1 M (enhancer) and 7 mL of fresh prewarmed media were added. Cells were incubated for 4 more days at 37 °C, and the conditioned medium (CM) was collected by pelleting the cells at 3000 rpm for 20 min at 4 °C. Subsequently, the CM supernatant was filtered with sterile 0.45 μm Sartorius Minisart high flow syringe filters. The filtered CM was dialyzed overnight against 20 mM TrisHCl (pH 7.5), 150 mM KCl, and 10 mM $CaCl_2$ at 4 °C to remove EDTA. The CM was concentrated to 10 mL using Amicon Ultra-15 10 kD concentrators at $1000 \times g$ and mixed every 20 min. The concentrated CM was incubated with TALON metal affinity resin beads (Takara) for 1 h at 4 °C (beads were pre-washed with 20 mM TrisHCl at pH 7.5, 125 mM NaCl, and 2 mM $CaCl_2$). The mix was centrifuged 5 min at $125 \times g$ and the beads were washed three times with 20 mM TrisHCl at pH 8.0, 125 mM NaCl, 2 mM $CaCl_2$, and 20 mM imidazole. The mini-PCDH15 target protein was eluted with the same buffer containing 200 mM imidazole. The protein was further concentrated to 1 mL and purified on a Superose-6 10/300 increase column (GE Healthcare) in 20 mM TrisHCl at pH 8.0, 150 mM KCl, and 5 mM $CaCl_2$. The proteins were concentrated, and the final concentration for quantitative analysis was determined by measuring absorbance using a Nano-Drop (Thermo Scientific).

Size exclusion chromatography coupled to multi-angle light scattering (SEC-MALS). Protein was analyzed by SEC-MALS using an ÄKTA Purifier system connected in series with a Wyatt miniDAWN TREOS system (Wyatt) with a Superose-6 10/300 increase column (GE Healthcare), using a flowrate of 0.5 mL/min at 4 °C. Proteins were monitored with both 280 nm absorbance and light scattering. The scattering data were subsequently converted into molecular weight using a rod-like model and normalized with the Astra software (Wyatt). SEC-MALS experiments were done with at least two biological replicates and accompanied by at least one duplicate. SDS-PAGE analyses were carried out after SEC-MALS to corroborate protein purity. SEC data was obtained using the Unicorn 5.31 software, and the MALS data was analyzed with the ASTRA 6.1 software. GraphPad Prism 9.5.0 software was used to generate the graphs.

## RNA extraction, cDNA production, reverse transcription, PCR amplification, and sequencing

Total RNA was collected from flash-frozen cochleas collected from 5–6 week old mice. Cochleas were stored at −80 °C prior to RNA extraction, which was performed using the Zymo Quick-RNA Microprep Kit (Zymo Research, #R1050) supplemented with a proteinase K digestion. RNA was reverse transcribed using Invitrogen's SuperScript IV VILO Master Mix with ezDNAse enzyme (Invitrogen, #11766050). PCDH15 cDNA was amplified using a forward primer in EC3 (5'-tcctgccttttgctagtctgc-3') and a reverse primer in EC9 (5'-gaagcgtgg-gatctctccag-3'). The expected product size from mini-PCDH15 version 4 transcripts is 388 bp while the expected product size for wild-type PCDH15 is 2,002 bp.

All PCR product bands were gel extracted using the DNA Gel Extraction Kit (Monarch, #T1020S). The purified PCR products were cloned into the pMiniT 2.0 vector backbone using the NEB PCR Cloning Kit (NEB, #E1202S), and transformed into competent cells. Three colonies were mini-prepped for each product and each was

subjected to Sanger sequencing using primers provided by the NEB PCR Cloning Kit.

### AAV round window membrane injection in neonatal mice

The RWM injections were performed under a stereomicroscope (Nikon SMZ1500). P0-1 pups were anesthetized using cryoanesthesia and kept on an ice pack during the procedure. Injections were done through the RWM as previously described[5]. Briefly, a small incision was made underneath the external ear. The incision was enlarged, and soft tissues were pushed apart to expose the bulla. The round window niche was localized visually. The viral vector solution was injected with a micropipette needle at a rate of 150 nL/min using a Nanoliter 2000 Injector (World Precision Instruments). The surgical incision was closed using a 7-0 Vycril surgical suture. Standard postoperative care was applied after the injection.

### ABR and DPOAE testing

ABRs and DPOAEs were recorded as previously described[64] using a custom acoustic system (Massachusetts Eye and Ear, Boston, MA, USA). Adult mice (from P21 to P90) were anesthetized with a ketamine/xylazine cocktail and placed on a 37 °C heating pad for the duration of the recording. Acoustic stimuli were delivered via an assembly consisting of two electrostatic drivers as sound sources and a miniature microphone at the end of a probe tube to measure sound pressure in situ. ABRs were recorded using three subdermal needle electrodes: reference electrode in the scalp between the ears, recording electrode just behind the pinna, and ground electrode in the back near the tail.

Tone-pip stimuli (5 ms with a 0.5 ms rise-fall time) at frequencies from 5.6–45.2 kHz were delivered in alternating polarity at 30 s$^{-1}$. The response was amplified (×10,000), band-pass filtered (0.3–3 kHz), and averaged (×512) with a PC-based data acquisition system using the Cochlear Function Test Suite software package (Massachusetts Eye and Ear, Boston, MA, USA). Sound levels were incremented in 5-dB steps, from -20 dB sound pressure level (SPL) up to either 80 or 129 dB. ABR Peak Analysis software (v1.1.1.9; Massachusetts Eye and Ear, Boston, MA, USA) was used to determine the ABR thresholds and to measure peak amplitudes. For traces with no detectable peaks, the amplitude was reported as 0 μV, and the animal was excluded from the peak measurements.

ABR thresholds were confirmed by visual examination as the lowest stimulus level in which a repeatable waveform could be observed. DPOAEs were recorded for primary tones (frequency ratio f2/f1 = 1.2, level ratio L1 = L2 + 10), where f2 varied from 5.6 to 45.2 kHz in half-octave steps. Primary tones were swept in 5-dB steps from 10 to 70 dB SPL (for f2). DPOAE threshold was determined from the average spectra as the f1 level required to produce a DPOAE of 5 dB SPL.

### FM1-43 loading in neonatal and adult cochlea

Neonatal P5-6 mice were anesthetized with cryoanesthesia and euthanized by decapitation. Inner ears were harvested and organ of Corti epithelia were acutely dissected in Leibovitz's L-15 cell culture medium, and were affixed to Cell-Tak-treated coverglass. Following tectorial membrane removal and medium aspiration, FM1-43 solution (2 μM in L-15) was applied to the tissue for 30–60 s, then quickly aspirated. The explant was then quickly rinsed with L-15, and the excess dye quenched by a 0.2-mM solution of SCAS (Biotium) in L-15.

Adult mice were anesthetized using Isoflurane, open drop method, and euthanized by cervical dislocation followed by decapitation. Mouse otic capsules were harvested and placed in Leibovitz's L-15 medium. Under a stereomicroscope, the cochlear bone was carefully peeled away with a 27-gauge needle, exposing apical and mid-apical regions of the cochlea. The tectorial membrane was pulled out to expose the sensory epithelium. FM1-43 solution (2 μM in L-15) was applied directly to the exposed epithelium and delivered throughout the cochlea by local perfusion through oval and round windows for

1 min at room temperature. Next, a 0.2-mM solution of SCAS was perfused through the oval and round windows. For both neonatal and adult mice, the organs of Corti were observed on an Olympus upright FV1000 confocal microscope equipped with a 60X 1.1 NA water-dipping objective lens.

### Single cell electrophysiology

Single cell electrophysiology was performed as described previously[54]. Neonatal P6 mice were anesthetized with cryoanesthesia and euthanized by decapitation. Cochlear dissections were performed, and then organs were cultured for an additional 4–11 days. Transduction currents were recorded from IHCs of the mid-basal area between P10-P17 using a whole-cell patch clamp to detect transduction currents and a stiff glass stimulating probe with a tip diameter of ~4 μm to match the shape of the IHC bundles. We focused our analysis on IHC bundles with minimal polarity defects. Bundles were displaced for 80 ms with 16 step displacements ranging from −175 to 1135 nm at 88-nm increments. For recordings, 1.5-mm glass pipettes were filled with an internal solution containing 137 mM CsCl, 5 mM EGTA, 10 mM HEPES, 2.5 mM Na$_2$-ATP, 0.1 mM CaCl$_2$, and 3.5 mM MgCl$_2$, and adjusted to pH 7.4 with CsOH, ~290 mmol/kg. The tissues were bathed in external solution containing 137 mM NaCl, 5.8 mM KCl, 0.7 mM NaH$_2$PO$_4$, 10 mM HEPES, 1.3 mM CaCl$_2$, 0.9 mM MgCl$_2$, 5.6 mM glucose, vitamins, and essential amino acids, adjusted to pH 7.4 with NaOH, ~310 mmol/kg. Cells were held at a − 80-mV potential, and a separate pipet flowed extracellular solution onto their apical surfaces. Data were analyzed with Clampfit, version 10.4.0.36.

### Immunofluorescence labeling of HEK293 cells and mouse cochleas

Neonatal mice were anesthetized with cryoanesthesia and euthanized by decapitation. Inner ears were harvested and organ of Corti explants were dissected in L-15 medium and fixed with 4% formaldehyde in HBSS for 1 h, washed three times with HBSS, and then blocked with 10% donkey serum for 2 h at room temperature. Samples were incubated in rabbit polyclonal anti-PCDH15 antibody (DC811, raised against peptide LSLKDNVDYWVLLDPVK corresponding to amino acids 80–96 in EC1; diluted 1:200 in 10% donkey serum) or anti-HA antibody, ChIP Grade (ab9110; diluted 1:500 in 10% donkey serum) for 24 hr at room temperature followed by several rinses in HBSS. Next, samples were incubated in blocking solution for 30 min and incubated overnight at room temperature with a donkey anti-rabbit immunoglobulin G (IgG) secondary antibody conjugated to Alexa Fluor 594 in a 1:500 dilution in blocking solution. To label hair bundle actin, we used phalloidin conjugated to Alexa Fluor 405 (1:20; Life Technologies).

Adult mice were anesthetized using Isoflurane, open drop method, and euthanized by cervical dislocation followed by decapitation. Cochleas were dissected in L-15 medium and immediately fixed with 4% formaldehyde in HBSS for 1 h at room temperature, washed with HBSS and transferred to fresh 10% EDTA for 2 days. After samples were fully decalcified, the organs of Corti were microdissected, blocked, and permeabilized with 10% donkey serum with 0.5% Triton X-100 for 1 h at room temperature. Samples were then stained with anti-HA antibody, ChIP Grade (ab9110; diluted 1:500 in 10% donkey serum) and incubated overnight at room temperature followed by several rinses in HBSS. Next, samples were incubated in blocking solution (10% donkey serum) for 30 min at room temperature and incubated overnight at room temperature with a donkey anti-rabbit IgG secondary antibody conjugated to Alexa Fluor 594 in a 1:500 dilution in blocking solution which included Alexa Fluor 405 phalloidin. After secondary antibody steps, samples were rinsed several times in HBSS and mounted on Colorfrost glass slides (Thermo Fisher Scientific) using Prolong Gold Antifade mounting medium (Thermo Fisher Scientific). The slides were kept in a horizontal position for drying in the dark for 24 h at room temperature before imaging with a

Nikon Ti2 inverted spinning disk confocal using Nikon Elements Acquisition Software AR 5.02 and using following objectives: a Plan Apo λ 100×/1.45 oil, a Plan Fluor λ 40×/1.3 oil, Plan Apo λ 60x/1.4 oil.

In experiments where transduction efficiency of AAV-PHP.B-optimized expression cassettes were evaluated via eGFP fluorescence, imaging was performed in apex, middle, and base regions with a Nikon Ti2 inverted spinning disk confocal using a Plan Fluor 40×/1.3 oil objective. The laser intensity was chosen based on the specimen with the strongest eGFP signal to prevent fluorescence saturation, and the same settings were then used for each image of a set.

For immunocytochemistry in HEK293 cells (ATCC, #CRL-1573), transfected cells were fixed with 4% formaldehyde in Hank's balanced salt solution (HBSS) for 1 h, washed three times with HBSS, and then blocked with 10% donkey serum for 2 h at room temperature. Rabbit polyclonal anti-PCDH15 antibody or anti-HA tag antibody was used. They were diluted 1:200 in 10% donkey serum and incubated 24 h at room temperature followed by several rinses in HBSS. Next, samples were incubated in blocking solution for 30 min and incubated overnight at room temperature with a donkey anti-rabbit immunoglobulin G (IgG) secondary antibody conjugated to Alexa Fluor 594 in a 1:500 dilution in blocking solution. After secondary antibody incubation, samples were rinsed several times in HBSS and mounted on a Colorfrost glass slide (Thermo Fisher Scientific) using Prolong Gold Antifade mounting medium (Thermo Fisher Scientific). Imaging was performed with an Olympus FluoView 1000 confocal microscope with a 60 × 1.42-NA oil-immersion objective.

## Conventional scanning electron microscopy

In neonatal and adult mice was performed as previously described[65]. Neonatal mice were anesthetized with cryoanesthesia and euthanized by decapitation. Adult mice were anesthetized using Isoflurane, open drop method, and euthanized by cervical dislocation followed by decapitation. Neonatal organ of Corti explants were dissected at P6 in L-15 medium and fixed with 2.5% glutaraldehyde in 0.1 M cacodylate buffer (pH 7.2), supplemented with 2-mM $CaCl_2$ for 1–2 h at room temperature. The samples were rinsed three times in sodium cacodylate buffer, 0.1 M, pH 7.4 for 10 min, and then briefly once in distilled water, dehydrated in an ascending series of ethanol, and critical-point dried from liquid $CO_2$ (Tousimis Autosamdri 815).

For adult mice, cochleas were quickly extracted from temporal bones and transferred into Leibovitz's L-15 solution. Under a stereomicroscope, a small hole was made in the apex of the cochlea using a 27 G needle. Cochleas were then pre-fixed immediately after dissection by immersion in 1% glutaraldehyde / 4% formaldehyde in 0.1 M cacodylate buffer (pH 7.2) supplemented with 2 mM $CaCl_2$, for 1 h at room temperature. After the prefixation step, the samples were postfixed with 2.5% glutaraldehyde in 0.1 M cacodylate buffer (pH 7.2) supplemented with 2 mM $CaCl_2$, for 1 h at room temperature, rinsed in 0.1 M cacodylate buffer (pH 7.2) and then rinsed in distilled water. The cochlear bone was carefully peeled out with a 27-gauge needle, then the organ of Corti was microdissected and the tectorial membrane was pulled out to expose the sensory epithelium. Next, the samples were immersed in a saturated aqueous solution of 1% osmium tetroxide for 1 h in the dark, washed once with water for 10 min, and postfixed with 1% tannic acid aqueous solution for 1 h in the dark. Finally, the samples were rinsed, dehydrated and critical point dried.

Samples were mounted on aluminum stubs with carbon conductive tabs and were sputter-coated (EMS 300 T dual-head sputter coater) with platinum to 5 nm and observed in a field-emission scanning electron microscope (Hitachi S-4700).

## Immunogold scanning electron microscopy in HEK293 cells and in mouse cochlea

Immunogold scanning electron microscopy was performed as previously described[64–66]. HEK293 cells (ATCC, #CRL-1573) on coverslips

were fixed with 4% formaldehyde in HBSS for 1 h at room temperature, then were rinsed 3 times in HBSS for 10 min. Cochleas were dissected and fixed as for conventional scanning electron microscopy (above). After fixation, samples (neonatal cochlea, adult cochlea or HEK293 cells) were blocked in 10% normal goat serum for 2 h at room temperature, incubated with primary antibodies for 24 h at room temperature, and rinsed for 10 min in HBSS three times. Rabbit polyclonal anti-PCDH15 antibody (DC811) was used (1:200 dilution) in 10% donkey serum to immunolabel PCDH15, and anti-HA tag antibody (Abcam, ab9110) in 10% donkey serum to label the HA tag (1:200 dilution). After rinsing, samples were blocked again in 10% normal goat serum for 30 min at room temperature and were incubated overnight at room temperature with secondary antibody solution, 12-nm Colloidal Gold AffiniPure Goat Anti-Rabbit IgG (Jackson ImmunoResearch, 111-205-144), 1:30 dilution in blocking solution. Following the secondary antibody application, the samples were rinsed in HBSS 3x for 10 min each. Finally, the samples were dehydrated, critical-point dried, mounted, sputter-coated with 3–5-nm palladium and observed in a field-emission scanning electron microscope with a backscattered electron detector, either a Hitachi S-4700 field-emission scanning electron microscope or a FEI Helios 660 FIB- scanning electron microscope.

## Quantification of confocal microscopy data and statistical analysis

Microscopy data analysis and quantification were done in the Fiji distribution of ImageJ v1.53. To measure eGFP fluorescence intensity at each cochlea region (apical, middle, basal) or FM1-43 fluorescence in the hair cells on confocal microscopy images, a ROI was generated inside the hair cell and the mean fluorescence intensity of each ROI was measured. The mean fluorescence intensity of an area outside the hair cells was considered as background and was subtracted from the values at the cell body for each image. To measure the fluorescence intensity of HA-tag labeling at the hair-cell bundles, an ROI was generated around hair bundles using a circular tool and the integrated fluorescence density of each ROI was measured and background subtracted.

Transduction efficiency of IHCs and OHCs was evaluated as previously described[5] using ImageJ. Hair cells were identified with phalloidin staining of bundles and transduced cells by positive HA-tag labeling. Control samples without AAV were used to correct for autofluorescence. Segments with dissection-related damage were removed from the analysis.

In quantification of rescue of morphology, a hair bundle was scored as "fully rescued" if morphologically similar to control hair bundles, with no sign of stereocilia loss and no sign of bundle fragmentation or disrupted polarity. A bundle that is "partially rescued" had ≥55% intact stereocilia per bundle, although some short and middle row stereocilia may have been missing and some bundle fragmentation or disrupted polarity may have been present. The bundle was considered "unrescued" if the hair bundle was severely disorganized, being fragmented or mispolarized, and lacked most short and middle row stereocilia with only some stereocilia of the tallest row remaining.

GraphPad Prism 7 software was used to generate the graphs and perform the statistical analysis. The results are shown as mean ± SEM or mean ± SD as indicated in figure legends. Randomization was used whenever possible.

## Statistics and reproducibility

All experiments, except where mentioned below, were reproduced in three independent experiments using independent samples.

The conventional scanning and immunogold electron microscopy presented in Fig. 2d, g, Fig. 3d, e, was reproduced in three independent experiments for the control and in four independent experiments for *Pcdh15^{fl/fl},Gfi1-Cre^{+/-}* and *Pcdh15^{fl/fl},Myo15-Cre^{+/-}* knockout mice. Experiments of Fig. 5c–f and Fig. 6d, were reproduced in three independent experiments for the control and in five independent experiments for

injected $Pcdh15^{fl/fl},Myo15\text{-}Cre^{+/-}$ mice, respectively. Experiments of Fig. 7a, b, g were reproduced in three independent experiments for the control and in five independent experiments for injected $Pcdh15^{R245X/R245X}$ mice, respectively.

The immunofluorescence analysis in Fig. 2f was reproduced in five independent experiments for the control and in three independent experiments for $Pcdh15^{fl/fl},Gfi1\text{-}Cre^{+/-}$ knockout mice. In Fig. 5a, Fig. 6a was reproduced in three independent experiments for the control and in twelve independent experiments for injected $Pcdh15^{fl/fl},Myo15\text{-}Cre^{+/-}$ knockout mice. Experiments of Fig. 7c, d were reproduced in four independent experiments for the control and in thirteen independent experiments for injected $Pcdh15^{R245X/R245X}$ mice, respectively. Experiments of Supplementary Fig. 5b were reproduced in four independent experiments for AAV-CBA-eGFP-WPRE-SV40-BGH and in five independent experiments for other AAVs.

FM1-43 loading experiments in Fig. 2e and Fig. 2f were reproduced in six independent experiments for the control and in four independent experiments for $Pcdh15^{fl/fl},Gfi1\text{-}Cre^{+/-}$ and $Pcdh15^{fl/fl},Myo15\text{-}Cre^{+/-}$ knockout mice respectively. Experiments of Fig. 5g were reproduced in five independent experiments for injected $Pcdh15^{fl/fl},Myo15\text{-}Cre^{+/-}$ knockout mice and of Fig. 8b in eight independent experiments for injected $Pcdh15^{R245X/R245X}$ mice.

Immunofluorescence and immunogold labeling of HEK293 and the cell aggregation assay in Supplementary Fig. 1, Supplementary Fig. 13 were reproduced in three independent experiments.

The genotyping for the $Pcdh15^{fl/fl}$ floxed allele (Supplementary Fig. 6b) was reproduced more than one hundred times.

Only samples with significant dissection-related damage were removed from the analysis. No other data were excluded.

All figures were assembled using Illustrator 2022 (Adobe, San Jose, CA, USA; v26.3.1.). The ABR data were sorted and calculated using Excel 2016 (Microsoft, Redmond, WA, USA; version 16.0.5378.1000) before the calculation of significance. GraphPad Prism 7 software (GraphPad Software Inc., Boston, MA, USA; version 7.04) was used to generate the graphs and perform the statistical analysis. The results are shown as mean ± SEM or mean ± SD as indicated in figure legends. Randomization was used whenever possible.

## Reporting summary
Further information on research design is available in the Nature Portfolio Reporting Summary linked to this article.

## Data availability
All data generated or analyzed during this study are included in this article and its supplementary information files. Source data are provided with this paper.

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

## Acknowledgements

We thank Drs. Christine Petit from Institut Pasteur and Ronna Hertzano from University of Maryland School of Medicine for providing *Myo15-Cre* mice. We thank Dr. Lin Gan from University of Rochester for providing *Gfi1-Cre* mice. We thank Dr. Joshua Sanes from Harvard University for providing *cdh2* knockout HEK293T cells. We appreciate the use of the Nikon Ti2 inverted spinning disk confocal at the Harvard Medical School MicRoN Microscopy Core and the Hitachi S-4700 scanning electron microscope at the Harvard Medical School Electron Microscopy Facility. We especially thank Bruce Derfler for laboratory management and Yaqiao Li for maintaining the mouse colony (Harvard Medical School). We thank Tyler Morris for help with cell culture, DNA transfection, and protein purification (Massachusetts Eye and Ear) and Drs. Chia-Lun Tsai and Brian Seed for providing access to SEC equipment (Center for Computational and Integrative Biology, Massachusetts General Hospi-tal). We also thank Dr. Jonathan Bird from University of Florida for assistance with Nano-SPD experiments. This work was supported by the National Institutes of Health (R01-DC016932 to D.P.C., R01-DC017166 to A.A.I., R01-DC015271 to M.S., and R01 DC020190-01 to A.A.I., D.P.C. and M.S.), by the Usher 1 F Collaborative (to D.P.C.), by the Bertarelli

Foundation (to D.P.C. and A.A.I.), by a Q-FASTR Award and a Blavatnik Therapeutics Challenge Award from Harvard Medical School (to D.P.C. and A.A.I.), and by NIGMS T32 Training in Genetics Fellowship T32 GM007748 (to K.T.B.).

## Author contributions

M.V.I. conceptualization, data acquisition, data analysis, data interpretation, visualization of in vivo and some in vitro experiments, manuscript writing; D.M.H. plasmid cloning, data acquisition, data analysis and data interpretation of ABR and behavioral experiments, RT-PCR experiments and some in vitro experiments, manuscript writing; A.J.K. data acquisition, analysis and interpretation of ABR and behavioral experiments; B.P. data acquisition, analysis of electrophysiology data; O.S. plasmid cloning, data acquisition, analysis and interpretation of NanoSPD experiments, manuscript writing; P.D. and J.B. plasmid cloning, protein expression, purification, data acquisition, analysis and interpretation of SEC-MALS experiments, manuscript writing; X.W. generation of Pcdh15fl/fl mouse; C.W.P. generation of R245X mouse; plasmid cloning; E.M.M. plasmid cloning, K.T.B. plasmid cloning, manuscript writing; C.G. data acquisition for ABR experiments; M.S. conceptualization, design of mini-PCDH15 constructs, manuscript writing; A.A.I. conceptualization, data acquisition, analysis and supervision of cell aggregation assay, immunogold labeling in HEK cells, NanoSPD and SEC-MALS experiments, manuscript writing; D.P.C. conceptualization, supervision, data interpretation, manuscript writing.

## Competing interests

D.P.C. is a cofounder of Skylark Bio. M.V.I. is a consultant for Skylark Bio. President and Fellows of Harvard College, Ohio State Innovation Foundation, Massachusetts Eye and Ear Infirmary have filed patent application "AAV vectors encoding mini-PCDH15 and uses thereof", inventors D.P.C., A.A.I., M.S., M.V.I. and C.W.P.; US (PCT/US2020/029968). The present disclosure provides isolated nucleic acids, vectors, and rAAV.9.PHP.B comprising a transgene encoding a mini-PCDH15, and methods of treating hearing loss using the same. The other authors declare no competing interests.
