## [Peer Review File · Nature Communications]

Mini-PCDH15 gene therapy rescues hearing in a mouse model of Usher syndrome type 1FREVIEWERS' COMMENTS

Reviewer #1 (Remarks to the Author):

Three mouse models with PCDH15 mutations, two conditional and one constitutive knock-out with premature stop, were studied for their hearing loss phenotype and the ability of mini-Pcdh15-AAVs to rescue hearing. These mini-Pcdh15-genes were designed according to previously solved structural data. While one mini-gene (V4) rescues hearing if applied early during development, the mini gene V7 even seems to be detrimental when injected into normal hearing mice. The paper advances our understanding of the protein Pcdh15, as well as providing a therapeutic perspective for patients with hearing loss due to mutations in PCDH15.

Significance

With respect to gene therapy for hearing impairment, several groups are testing different strategies for the transfer of genes (often long cDNAs) to sensory cells of the inner ear. The comparison of different approaches shall serve to find the best strategy for each condition. The work presented here provides a comprehensive study using gene supplementation with mini-genes in adeno-associated viral vectors. The test of different promoter and mRNA stabilizing sequences will be useful for the design of gene supplementation strategies for other conditions. With respect to related fields, this work demonstrates that based on knowledge about structure-function relationship, it is well possible to design a functional mini gene that might be used for therapeutic purposes in the future.

The data are well suited to support the claims. The read-out for the success of the gene therapy is elaborate and state-of-the-art. All experimental procedures are described in very detail.

Major comments – requiring revision/discussion:

What is the latest time point where the AAV injection will rescue hearing in any of the mouse models tested? From the literature, it can be assumed that it is presumably up to P5 in mice, corresponding to the 19th week of gestation in humans (Hastings and Brigande, 2020). Please state this limitation, at least for those patients with congenital onset of hearing loss.

Fig 7e and 8 c: The number of OHCs that take up FM1-43 seems to be higher than the number of anti-HA labelled OHCs. How could this discrepancy be explained?

Line 482f: "Delivery of a functional coding sequence at P0 [to Pcdh15R245X mice] was not sufficient to restore bundle morphology,..." From the data presented, this finding is not so obvious. I recommend to include a quantification of hair bundle morphology similar as in Fig 4a'.

Minor comments

The results section starts with Fig 1 and supplementary figures, which is ok, but the next main figure mentioned is Fig 6. I suggest to re-arrange the figures to match with the text.

Line 40: "More recently, mini-otoferlin genes have been used to probe the role of 40 the protein in hair-cell endocytosis in the cochlea Ref13"
The reference is not correct. Mini-Otoferlin has been employed in Tertrais et al, 2019, while Rankovic et al (2020) used full-length otoferlin.

Line 96 ff

"Although these models lack bound Ca²⁺, they were similar to high-resolution X-ray crystal structures of PCDH15 regions that have been solved previously 17-19, 23, 26, 37,

suggesting that the mini-PCDH15s would fold properly and make bona-fide EC-EC junctions.”

In order to better understand this sentence, it would be helpful if the way how AlphaFold2 predicts structures would quickly be explained (i.e., it can provide structures in a template-free manner).

Fig 1: In order to more intuitively connect panels a and b, I suggest to mirror the drawing in panel a, such that Protocadherin is on the right side.

Fig 1, Figure legend: Although obvious to people from the field, it might be helpful for a broader readership to indicate that in b a homodimer is displayed, together with the connection to two EC repeats of Cadherin-23.

Fig 1c: it is unclear to me which structure represents the AlphaFold2 with and the one without template.

Are both displayed? Or only one, since they are identical anyway?

Lines 121-122, “Together, these assays suggest that at least some mini-PCDH15s will be functional in hair cells.”

Since the true functional assays come later, I suggest to restrict the statement here to trafficking/surface expression and Cdh23 interaction.

Line 368: “To investigate the ability of mini-PCDH15-V4 rescue the function...” – A “to” will increase readability.

Line 390: “no current was detected in Pcdh15R245X/R245X mice (Fig. 8d, e).”

I did not find the data for untreated Pcdh15R245X/R245X mice in the figure.

Line 457: “mini-PCDH14-V4” – please correct.

Line 459: “Interestingly, even though the expression of eGFP with AAV9-PHP.B was relatively high in the base, the rescue level was highest in the apical and middle regions of the cochlea.”

It is not clear to me what you find so interesting about this – did you expect a rescue which is in the base as good as in the middle and apex?

Fig 2:

The labels within the graphs in Fig 2d-e are hard to read, I would recommend to use bold lettering or a black shade behind the white letters.

Fig 3, figure legend: “c” is missing in the legend, correct also “d” and “e”. For “f” and “g” you describe in the legend “wild-type” mice as controls, but the panel legend says Pcdh15fl/fl. The same applies to the legend of Supp Fig 8 legend and to the results line 242.

Line 279 and Fig 4, Figure legend: please also replace “wild-type” with “Pcdh15fl/fl”.

Fig 6: Please indicate the number of animals in panel e (in the legend).

Fig 7, panel e legend: “...treated mice at P6” – this could be misread as if the mice were treated at P6. Please rephrase.

Fig 8: legend to panel c, please substitute “wild-type control mice” with Pcdh15R245x/+ control mice. Legend to panel e: please indicate that the homozygous R245X mice have

been treated with AAV-mini-PCDH15-4. Panel g, please indicate the number of animals in each group.

Supplementary Figure 1: A scale bar for panel c would be helpful.

Supplementary Figure 3: Check if CDH23 + mini-PCDH15V1 is really not significantly different to the positive control (blue “ns”).

Supplementary Figure 5, panel a: It would be helpful if the abbreviation CMV in the sketch would be more clearly labelled as CMV enhancer or CMV promoter. The same applies to Fig 3 a.

Supplementary Figure 6, panel b: Two PCR protocols are described in the legend, but it is unclear which PCR is depicted in the image.

Supplementary Figure 10: Which Cre mouse line was used here?

Supplementary Figure 11: Please indicate the numbers of animals.

Supplementary Figure 13: Please indicate the length of the scale bars.

Reviewer #2 (Remarks to the Author):

The manuscript by Ivanchenko et al. describes the development of a novel gene therapy strategy for Usher syndrome 1F. The authors explore miniaturized variants of PCDH15 that can be fitted into a single AAV vector for their capability to restore hearing function in mouse models of USH1F. Their comprehensive study covers the entire process from in silico variant design, over in vitro testing of recombinant variants up to in vivo application in newly generated mutant mice for rescue of the USH1F phenotype. After in silico design of eight shortened PCDH15 variants, the authors tested them for correct surface expression, CDH23 interaction and dimerization in vitro. Three of the variants were chosen for packing into recombinant AAV vectors for application in vivo. For this part, Ivanchenko et al developed and characterized a novel conditional Pcdh15 mouse mutant crossbred to two different Cre lines for Pcdh15 ablation at two different time points in hair cell development and hence distinct severity. Hearing loss and inner and outer hair cell morphology was very well characterized. Gene therapy was applied initially to the milder affected line and for the most successful Mini-PCDH15 variant extended to the more severely affected conditional mouse line and furthermore to an established constitutional Pcdh15R245X mouse line. The datasets presented are highly conclusive with respect to time points of treatment and analysis and parameters investigated. The data is presented in eight main and 13 supplemental figures of high quality. The manuscript is very well structured and overall well written. One aspect that the authors need to work on is the perspective for clinical translation: will there be a therapeutic window in humans where hair cell degeneration likely prevails at birth. There are some issues with referencing that the authors want to take care of: generally, this is a bit of a “Boston-centric” referencing and generally original publications of other labs should be preferred over reviews, e.g. for dual-AAV and overload approaches. Reference 13 meant to point to the mini-Otof study of the Petit lab incorrectly points to the AAV-overload study of the Moser lab. Overall the manuscript is of very high interest to the field of hearing research and of excellent experimental quality. One aspect the Usher field would anxiously want to know is whether vestibular and visual deficits exist and are amenable to the same therapeutic approach. From my perspective this, however, is not strictly required for publication of this comprehensive preclinical study on hearing in USH1F

mouse models I wish to congratulate the authors on.

Major remarks:

1. The authors emphasize the importance of the linker regions between the EC domains and maintenance of their functionality in the Mini-Pcdh15 construct. Detailed protein sequence information of the variants is missing and needs to be included. Only for V4, some sequence information is hidden in supplemental figure 12c. Favorable would be full plasmid sequences for all constructs included e.g. by deposition at a public data repository.
2. Is the retinal photoreceptor integrity of the Pcdh15^{fl/fl} Gfi1Cre/+ mice impaired? Does gene therapy using Mini-PCDH15V4 rescue a possible phenotype? Did you observe vestibular impairment? Were vestibular hair cells also targeted by the gene therapy applied? Was a possible phenotype rescued by your gene therapy approach?

Minor remarks:

1. Please provide all statistical tests applied and exact p values with the respective data.
2. Color codes in figure 3 d to g and 4a are hard to discriminate, e.g. due to size and similar colors. Please increase size and use a color scheme that is adapted to color blindness.
3. In Figure 3d', have data for V4 and V4-HA been merged into one group?
4. Labelling in Figure 5b is not readable, please increase in size or restructure.
5. In Figures 5c, 7e and 4a' efficiency data for the V4 gene therapy applied to the 3 models is provided. However, the data representation differs between the three datasets not allowing direct comparison. Does the model used have an impact on fluorescence intensity, transduction efficiency or rescue rate, respectively? Since the models differ in effect size in ABR data, how does this translate to quantification of the morphological appearance? Are the data indicative for possible limitations of a post natal gene therapy in future clinical trials?
6. In Supplemental Figure 4a you state that all constructs were detected as dimers. Elution profile and MALS of full length PCDH15 indicate a higher molecular weight structure that is not detected for the shortened versions. Is the difference observed only due to monomer length (in that case, please indicate in the figure) or are there more complex interactions formed that were not restored for the Mini-PCDH15 variants?
7. Throughout the manuscript, Pcdh15^{fl/fl} mice were labelled as "wild type mice". Please change the term to "control mice".

Reviewer #3 (Remarks to the Author):

In this study, the authors examined whether inner ear gene therapy could be applied to three different mutant PCDH15 mouse models. Since PCDH15 is a large gene, they engineered several mini-PCDH15s and tested them to see if these mini versions of PCDH15 could restore auditory function in three different PCDH15 mouse mutants. The application of "mini-PCDH15" PCDH15 mutants is novel. The experiments were well designed and well controlled. The manuscript was well-written. My comments are listed below.

1. The testing of different promoter/poly(A)/WPRE is interesting. Since the inclusion of WPRE seems to effect the eGFP intensity, and not transduction rate, would its inclusion be helpful to improve the auditory function in the Gfi1-Cre and the R245X mutants?
2. My main criticism with this manuscript is the claim that there is "robust rescued hearing" with the Gfi1-Cre and R245X/R245X mouse models. The ABR thresholds for both of these mouse models are far from normal. I would suggest the authors to simply state what the actual ABR thresholds are for these mouse models and let the readers decide whether this recovery is robust or not.

REVIEWERS' COMMENTS

Reviewer #1 (Remarks to the Author):

Three mouse models with PCDH15 mutations, two conditional and one constitutive knock-out with premature stop, were studied for their hearing loss phenotype and the ability of mini-Pcdh15-AAVs to rescue hearing. These mini-Pcdh15-genes were designed according to previously solved structural data. While one mini-gene (V4) rescues hearing if applied early during development, the mini gene V7 even seems to be detrimental when injected into normal hearing mice. The paper advances our understanding of the protein Pcdh15, as well as providing a therapeutic perspective for patients with hearing loss due to mutations in PCDH15.

Significance

With respect to gene therapy for hearing impairment, several groups are testing different strategies for the transfer of genes (often long cDNAs) to sensory cells of the inner ear. The comparison of different approaches shall serve to find the best strategy for each condition. The work presented here provides a comprehensive study using gene supplementation with mini-genes in adeno-associated viral vectors. The test of different promotor and mRNA stabilizing sequences will be useful for the design of gene supplementation strategies for other conditions. With respect to related fields, this work demonstrates that based on knowledge about structure-function relationship, it is well possible to design a functional mini gene that might be used for therapeutic purposes in the future.

The data are well suited to support the claims. The read-out for the success of the gene therapy is elaborate and state-of-the-art. All experimental procedures are described in very detail.

Major comments – requiring revision/discussion:

What is the latest time point where the AAV injection will rescue hearing in any of the mouse models tested? From the literature, it can be assumed that it is presumably up to P5 in mice, corresponding to the 19th week of gestation in humans (Hastings and Brigande, 2020). Please state this limitation, at least for those patients with congenital onset of hearing loss.

For the rescue experiments in this study using conditional or constitutive PCDH15 knockout mouse models, all injections were performed at the age of P0-P1. We agree with the reviewer's criticism that it's important to note that the immature, non-hearing inner ear of neonatal mice at P0-P1 is not a perfect model for the functionally mature, hearing inner ear of neonatal humans. The window of therapeutic efficacy for humans may close one week prior to hearing onset at 18 weeks gestational age, based on the postnatal window of efficacy observed in mice (Hastings ML, Brigande JV, 2020).

We added following to the discussion:

“While AAV injection in the mouse inner ear may provide insights into potential therapies for congenital hearing loss, it may not necessarily translate directly to humans. The human inner ear develops on a different timescale relative to birth than that of mice, and there may be differences in the optimal timing, dosage, and delivery method of AAV vectors. The window of therapeutic efficacy for humans may close one week prior to hearing onset at 18 weeks gestational age, based on the postnatal window of efficacy observed in mice (Hastings ML, Brigande JV, 2020), and intervention beyond this critical period may not be as effective in restoring hearing. However, some mild mutations (e.g., R134G, G262D, and V528D) are not deleterious to balance and vision

and are associated with late-onset hearing loss, which can potentially be treated (Ahmed et al., 2003; Doucette et al., 2009). In addition, vestibular hair cells, in particular, may be more responsive to therapy, as they continue to differentiate even after birth in mice (Hastings ML, Brigande JV, 2020). Thus, early intervention targeting a subset of vestibular hair cells may be possible. However, the most promising initial therapeutic target for Usher 1F is progressive blindness. Fortunately, the progression of blindness in Usher 1F occurs gradually over several decades, providing a sizable window of opportunity for treatment (Sethna, S. et al, 2021; Hartong, D.T. et al, 2006; Geleoc, G.G.S. & El-Amraoui, 2020; Mathur, P. & Yang, J., 2015)”.

Fig 7e and 8 c: The number of OHCs that take up FM1-43 seems to be higher than the number of anti-HA labelled OHCs. How could this discrepancy be explained?

In treated animals at P6 the proportion of FM1-43 positive OHCs is $81.1 \pm 2.0\%$ and the transduction efficiency counted as number of HA-tag positive OHCs is: $78.4 \pm 4.2\%$ in the apex, $79.4 \pm 3.5\%$ in the middle $73.3 \pm 4.0\%$ in the base. We can speculate that the FM1-43 test is more sensitive as only a few functional transduction channels are needed for detectable FM1-43 loading. In a small fraction of hair cells some cells could have some intact tip links, whereas the PCDH15 protein was not detected by immunofluorescence, resulting in minor differences that were not statistically significant.

Line 482f:” Delivery of a functional coding sequence at P0 [to Pcdh15R245X mice] was not sufficient to restore bundle morphology,...” From the data presented, this finding is not so obvious. I recommend to include a quantification of hair bundle morphology similar as in Fig 4a’.

We now present quantification of the rescue of the hair bundle morphology in treated R245X mice in Fig. 7f.

Minor comments

The results section starts with Fig 1 and supplementary figures, which is ok, but the next main figure mentioned is Fig 6. I suggest to re-arrange the figures to match with the text.

As per the reviewer's suggestion, we have re-arranged Fig. 6 and renamed it as Fig. 2 to match the corresponding text.

Line 40: “More recently, mini-otoflerin genes have been used to probe the role of 40 the protein in hair-cell endocytosis in the cochlea Ref13”. The reference is not correct. Mini-Otoferlin has been employed in Tertrais et al, 2019, while Rankovic et al (2020) used full-length otoferlin.

We have made the necessary updates to the reference and included the correct one (Tertrais M. et al., 2019). However, we kept the reference to Rankovic et al. (2020) since the study, besides using full-length otoferlin, tested a truncated otoferlin named in that study as “mini-Otoferlin.”

Line 96 ff

“Although these models lack bound Ca^{2+} , they were similar to high-resolution X-ray crystal structures of PCDH15 regions that have been solved previously 17-19, 23, 26, 37, suggesting that the mini-PCDH15s would fold properly and make bona-fide EC-EC junctions.” In order to better understand this sentence, it would be helpful if the way how AlphaFold2 predicts structures would quickly be explained (i.e., it can provide structures in a template-free manner).

We added the following sentence to the text:

“AlphaFold2 uses a deep machine learning algorithm, incorporating physical and biological knowledge about protein structure and multi-sequence alignments, to predict protein structures without relying on previously solved structures or structural templates.”

Fig 1: In order to more intuitively connect panels a and b, I suggest to mirror the drawing in panel a, such that Protocadherin is on the right side.

Good idea. We mirrored the drawing in Fig. 1a, and PCDH15 is now shown on the right side.

Fig 1, Figure legend: Although obvious to people from the field, it might be helpful for a broader readership to indicate that in b a homodimer is displayed, together with the connection to two EC repeats of Cadherin-23.

We updated the figure and legend.

Fig 1c: it is unclear to me which structure represents the AlphaFold2 with and the one without template. Are both displayed? Or only one, since they are identical anyway?

Both structural predictions made with and without known structural templates resulted in high confidence models that were identical to each other. However, in Fig. 1c, the "without templates" model was displayed, although both models were nearly identical. To avoid confusion for the reader, we have modified Figure 1c by keeping only "without template models". Additionally, we have clarified in both the text and the figure legend that both models are nearly identical.

Lines 121-122, “Together, these assays suggest that at least some mini-PCDH15s will be functional in hair cells.” Since the true functional assays come later, I suggest to restrict the statement here to trafficking/surface expression and Cdh23 interaction.

We modified the statement as reviewer suggested. Now it is: “Together, these assays suggest that all mini-PCDH15s go to the cell surface and they bind to CDH23 as well as full-length PCDH15”.

Line 368: “To investigate the ability of mini-PCDH15-V4 rescue the function...” – A “to” will increase readability.

Thanks for catching that. We now write “To investigate the ability of mini-PCDH15-V4 to rescue the function...”

Line 390: “no current was detected in *Pcdh15*^{R245X/R245X} mice (Fig. 8d, e).” I did not find the data for untreated *Pcdh15*^{R245X/R245X} mice in the figure.

We present data from *Pcdh15*^{R245X/R245X} mice in Supplementary figure 15

Line 457: “mini-PCDH14-V4” – please correct.

Typo has been corrected.

Line 459: “Interestingly, even though the expression of eGFP with AAV9-PHP.B was relatively high in the base, the rescue level was highest in the apical and middle regions of the cochlea.” It is not clear to me what you find so interesting about this – did you expect a rescue which is in the base as good as in the middle and apex?

Since we detected transduction with PHP.B-CMV-eGFP-BGH of hair cells in the base that was as good as in the apex and middle, we expected to see functional rescue with PHP.B-CMV-miniPCDH15.v4-BGH that was as good in the base as in the middle and apex.

We changed the sentence to “Interestingly, even though the expression of eGFP with AAV9-PHP.B was relatively high in the base, the rescue level was highest in the apical and middle regions of the cochlea and significantly lower in the base, suggesting that the efficiency of eGFP expression did not directly correlate with the efficiency of PCDH15 expression”.

Fig 2: The labels within the graphs in Fig 2d-e are hard to read, I would recommend to use bold lettering or a black shade behind the white letters.

Done

Fig 3, figure legend: “c” is missing in the legend, correct also “d” and “e”. For “f” and “g” you describe in the legend “wild-type” mice as controls, but the panel legend says *Pcdh15*fl/fl. The same applies to the legend of Supp Fig 8 legend and to the results line 242.

Done

Line 279 and Fig 4, Figure legend: please also replace “wild-type” with “*Pcdh15*fl/fl”.

Done

Fig 6: Please indicate the number of animals in panel e (in the legend).

Done

Fig 7, panel e legend: “...treated mice at P6” – this could be misread as if the mice were treated at P6. Please rephrase.

We corrected the wording

Fig 8: legend to panel c, please substitute “wild-type control mice” with *Pcdh15*^{R245X/+} control mice. Legend to panel e: please indicate that the homozygous R245X mice have been treated with AAV-mini-PCDH15-4. Panel g, please indicate the number of animals in each group.

Done

Supplementary Figure 1: A scale bar for panel c would be helpful.

Done

Supplementary Figure 3: Check if CDH23 + mini-PCDH15V1 is really not significantly different to the positive control (blue “ns”).

We have confirmed that CDH23 + mini-PCDH15-V1 is really not significantly different to the positive control ($p=0.09$). The result is likely due to a very low number of observations (measurements) for mini-PCDH15-V1.

Supplementary Figure 5, panel a: It would be helpful if the abbreviation CMV in the sketch would be more clearly labelled as CMV enhancer or CMV promoter. The same applies to Fig 3 a.

Done

Supplementary Figure 6, panel b: Two PCR protocols are described in the legend, but it is unclear which PCR is depicted in the image.

We have made revisions to the figure to ensure clarity for readers.

Supplementary Figure 10: Which Cre mouse line was used here?

Pcdh15^{fl/fl}, *Myo15-Cre*^{+/-}, as displayed in the figure. We have added this information to the legend.

Supplementary Figure 11: Please indicate the numbers of animals.

Done

Supplementary Figure 13: Please indicate the length of the scale bars.

Done

Reviewer #2 (Remarks to the Author):

The manuscript by Ivanchenko et al. describes the development of a novel gene therapy strategy for Usher syndrome 1F. The authors explore miniaturized variants of PCDH15 that can be fitted into a single AAV vector for their capability to restore hearing function in mouse models of USH1F. Their comprehensive study covers the entire process from in silico variant design, over in vitro testing of recombinant variants up to in vivo application in newly generated mutant mice for rescue of the USH1F phenotype. After in silico design of eight shortened PCDH15 variants, the authors tested them for correct surface expression, CDH23 interaction and dimerization in vitro. Three of the variants were chosen for packing into recombinant AAV vectors for application in vivo. For this part, Ivanchenko et al developed and characterized a novel conditional Pcdh15 mouse mutant crossbred to two different Cre lines for Pcdh15 ablation at two different time points in hair cell development and hence distinct severity. Hearing loss and inner and outer hair cell morphology was very well characterized. Gene therapy was applied initially to the milder affected line and for the most successful Mini-PCDH15 variant extended to the more severely affected conditional mouse line and furthermore to an established constitutional Pcdh15R245X mouse line. The datasets presented are highly conclusive with respect to time points of treatment and analysis and parameters investigated. The data is presented in eight main and 13 supplemental figures of high quality. The manuscript is very well structured and overall well written. One aspect that the authors need to work on is the perspective for clinical translation: will there be a therapeutic window in humans where hair cell degeneration likely prevails at birth. There are some issues with referencing that the authors want to take care of: generally, this is a bit of a “Boston-centric” referencing and generally original publications of other labs should be preferred over reviews, e.g. for dual-AAV and overload approaches. Reference 13 meant to point to the mini-Otof study of the Petit lab incorrectly points to the AAV-overload study of the Moser lab. Overall the manuscript is of very high interest to the field of hearing research and of excellent experimental quality. One aspect the Usher field would anxiously want to know is whether vestibular and visual deficits exist and are amenable to the same therapeutic approach. From my perspective this, however, is not strictly required for publication of this comprehensive preclinical study on hearing in USH1F mouse models I wish to congratulate the authors on.

We understand the importance of references in providing credibility to research work, and apologize for not including all of them in our paper. However, due to the limitations on allowable reference number, in some cases we had to rely on citing reviews. We have also made the necessary updates to the reference regarding the mini-Otof study and ensured that the correct one (Tertrais M. et al., 2019) has been now included. We kept the reference to Rankovic et al., 2020 since that study, besides using full-length otoferlin, also tested a truncated otoferlin referred to as “mini-Otoferlin”

Major remarks:

1. The authors emphasize the importance of the linker regions between the EC domains and maintenance of their functionality in the Mini-Pcdh15 construct. Detailed protein sequence information of the variants is missing and needs to be included. Only for V4, some sequence information is hidden in supplemental figure 12c. Favorable would be full plasmid sequences for all constructs included e.g. by deposition at a public data repository.

We included protein sequence information of the variants in a new Supplementary Table 1. We also deposited to Addgene (deposit 82408) the AAV plasmids used in mouse rescue experiments. They will be accessible to the public once the manuscript is published.

2. Is the retinal photoreceptor integrity of the *Pcdh15*^{fl/fl} *Gfi1*^{Cre/+} mice impaired?

We have not examined whether *Pcdh15*^{fl/fl}, *Gfi1*^{Cre+} mice exhibit a retinal phenotype. *Gfi1* is a transcription factor that plays a critical role in the development and maintenance of hematopoiesis and the inner ear; it is expressed in developing retinal ganglion cells but not in photoreceptors, where *Pcdh15* is expressed. Therefore, we doubt that *Pcdh15*^{fl/fl}, *Gfi1*^{Cre+} mice would show any retinal phenotype.

Does gene therapy using Mini-PCDH15V4 rescue a possible phenotype?

Our belief is that mini-PCDH15 therapies, which have been shown to preserve hearing in mouse models, are also likely to prevent blindness. Currently, we are investigating the efficacy of mini-PCDH15-V4 in the zebrafish *Pch15b* knockout retina, which does exhibit vision deficits. We have indeed observed rescue in zebrafish retina, but this is a separate line of research.

Did you observe vestibular impairment? Were vestibular hair cells also targeted by the gene therapy applied? Was a possible phenotype rescued by your gene therapy approach?

We did not observe any vestibular impairment in either *Gfi1-Cre* or *Myo15-Cre* conditional knockout mice using standard behavioral tests. We speculate that mice with a later deletion of PCDH15 learn to associate remaining vestibular sensation with proprioceptive sensation, and then can rely solely on proprioception after the PCDH15 is gone or *Gfi1-Cre* and *Myo15-Cre* are not activated as robustly in vestibular hair cells. For a future study, we will test vestibular function directly with VsEP recording.

In *R245X* constitutive knockout mice, however, a severe vestibular phenotype was evident behaviorally. In a separate study, we are carefully evaluating the effectiveness of mini-PCDH15-V4 in treating vestibular deficits. The preliminary results have shown a robust rescue of vestibular impairment.

Minor remarks:

1. Please provide all statistical tests applied and exact p values with the respective data.

Done

2. Color codes in figure 3 d to g and 4a are hard to discriminate, e.g. due to size and similar colors. Please increase size and use a color scheme that is adapted to color blindness.

We apologize for the inconvenience caused by the difficulty in discriminating the color codes. We understand the importance of having a color scheme that is easy to distinguish, particularly for individuals with color blindness. We appreciate the suggestion and made necessary changes.

3. In Figure 3d', have data for V4 and V4-HA been merged into one group?

We updated Fig. 3d (now Fig.4d), so it now presents both mini-PCDH15-V4 and mini-PCDH15-V4-HA data.

4. Labelling in Figure 5b is not readable, please increase in size or restructure.

We restructured Fig. 5 (now Fig. 6). We moved panel b with the predicted structure of the HA peptide added to the N-terminus of PCDH15 to Supplementary Figure 13 and increased the size.

5. In Figures 5c, 7e and 4a' efficiency data for the V4 gene therapy applied to the 3 models is provided. However, the data representation differs between the three datasets not allowing direct comparison. Does the model used have an impact on fluorescence intensity, transduction efficiency or rescue rate, respectively? Since the models differ in effect size in ABR data, how does this translate to quantification of the morphological appearance? Are the data indicative for possible limitations of a post natal gene therapy in future clinical trials?

We appreciate the reviewer's questions, and as suggested, we have restructured our quantification analysis to allow model comparison. We have included data on transduction efficiency and rescue rate for both conditional and constitutive models. Specifically, we used the late deletion *Pcdh15^{fl/fl}*, *Myo15-Cre* mouse as a platform to test different mini-PCDH15 versions to understand their relative efficacy. Our findings show that the best version (version 4) in the *Myo15-Cre* mouse was able to fully rescue tip links and preserve normal length of stereocilia. Transduction efficiency represents successful delivery and expression of the vector, while the rescue rate shows that the expressed protein rescues the morphology. We quantified fluorescence intensity to determine the number of PCDH15 molecules on the bundle, and we found that despite 60% transduction efficiency, cells at the base had 50% fewer PCDH15 molecules per bundle, which could be a possible reason for low rescue in the base.

In R245X mice, despite the rescue of tip links and stereocilia length, we observed a high rate of partial rescue characterized by the presence of split and misspolarized bundles. This suggests that although functional PCDH15 was present, the delivery of the vector was too late to fully rescue the bundle architecture, and some changes may be irreversible resulting in limited ABR rescue. Our findings suggest that intervention beyond a critical period may not be as effective in restoring hearing.

6. In Supplemental Figure 4a you state that all constructs were detected as dimers. Elution profile and MALS of full length PCDH15 indicate a higher molecular weight structure that is not detected for the shortened versions. Is the difference observed only due to monomer length (in that case, please indicate in the figure) or are there more complex interactions formed that were not restored for the Mini-PCDH15 variants?

We thank the reviewer for bringing to our attention the SEC-MALS and dimerization plots. The presented experimental molecular weight for the mouse WT PCDH15 EC1-MAD12 protein (blue curve, elution peak at 12 mL) reported at MW = 474 kDa includes glycosylation and is higher than

the non-glycosylated theoretical MW of WT PCDH15 EC1-MAD12 without its signal peptide (theoretical dimeric MW = 305 kDa, blue dashed line – now incorporated in the figure as a reference). We agree with the reviewer’s comments that the WT PCDH15 (blue) might display higher molecular weight structures in solution, as indicated by the experimental SEC-MALS data. In addition, we think that these higher molecular weight structures are also detected for the shortened versions (see shoulder peaks at ~ 12 mL in panels Supplementary Fig. 4a and Supplementary Fig. 4b, red asterisk, now described in the figures and the legend). Importantly, these shortened variants show mostly dimeric molecules at elution peaks of ~ 14 mL, while a mutant mini-v4 monomer elutes at ~ 15 mL (pink curve, Supplementary Fig. 4b).

We used the same cDNA of WT PCDH15 from our previous PNAS 2020 study to obtain our current SEC-MALS data. Our current SDS-PAGE analyses suggest that the WT protein migrates with an apparent molecular weight of ~200 to ~220 kDa (rather than the theoretical monomeric value of 152 kDa, see gel), consistent with Choudhary et al., PNAS 2020, Fig. S5.

Given the confounding effect of glycosylation and other uncertainties, we prefer to refrain from speculating about higher molecular weight structures for the WT PCDH15. We have updated figure labels and figure caption to provide further guidance on the interpretation of the data.

Redacted

7. Throughout the manuscript, *Pcdh15*^{fl/fl} mice were labelled as “wild type mice”. Please change the term to “control mice”.

Yes, that should be clarified. We have replaced “wild type mice” with “control mice” throughout the text.

Reviewer #3 (Remarks to the Author):

In this study, the authors examined whether inner ear gene therapy could be applied to three different mutant PCDH15 mouse models. Since PCDH15 is a large gene, they engineered several mini-PCDH15s and tested them to see if these mini versions of PCDH15 could restore auditory function in three different PCDH15 mouse mutants. The application of “mini-PCDH15” PCDH15 mutants is novel. The experiments were well designed and well controlled. The manuscript was well-written. My comments are listed below.

1. The testing of different promoter/poly(A)/WPRE is interesting. Since the inclusion of WPRE seems to effect the eGFP intensity, and not transduction rate, would its inclusion be helpful to improve the auditory function in the *Gfi1-Cre* and the *R245X* mutants?

The addition of a WPRE element to a gene therapy vector improves transgene expression by enhancing mRNA nuclear export and stability, potentially leading to a beneficial effect on hearing rescue in *Gfi1-Cre* and *R245X* mutants. However, even though we made shortened *Pcdh15* versions, addition of the promoter and BGH polyA brings our plasmids close to 5 kb in size, unfortunately leaving no room to add the WPRE element (Fig. 4a).

2. My main criticism with this manuscript is the claim that there is “robust rescued hearing” with the *Gfi1-Cre* and *R245X/R245X* mouse models. The ABR thresholds for both of these mouse models are far from normal. I would suggest the authors to simply state what the actual ABR thresholds are for these mouse models and let the readers decide whether this recovery is robust or not.

We appreciate the suggestion and in the revised manuscript we removed “robust”, allowing readers to draw their own conclusions.